# Relict Landscape Evolution and Fault Reactivation in the eastern Tianshan: Insights from the Harlik Mountains

Zihao Zhao[1], Tianyi Shen[2, *], Guocan Wang[1, 2, *], Peter van der Beek[3], Yabo Zhou[4], Cheng Ma[2]

[1]Institute of Geological Survey, China University of Geosciences, Wuhan, 430074, China
[2]Center for Global Tectonics, School of Earth Sciences, China University of Geosciences, Wuhan, 430074, China
[3]Institute of Geosciences, University of Potsdam, Potsdam, 14476, Germany
[4]PowerChina Huadong Engineering Corporation Limited, Hangzhou, 310012, China

*Correspondence to*: Tianyi Shen (shenty@cug.edu.cn) and Guocan Wang (wgcan@cug.edu.cn)

**Abstract.** Relict low-relief surfaces, formed during tectonically quiescent periods and later modified by factors such as increased tectonic activity, are prevalent within the active mountain ranges of Central Asia. However, their formation, preservation, and subsequent evolution within the Mesozoic-Cenozoic tectonic framework remain poorly understood. This study examines the low-relief surfaces of the Harlik Mountains, located in the easternmost Tianshan, integrating digital terrain analysis, fluvial geomorphic analysis, structural geology, and low-temperature thermochronology to reconstruct their long-term geomorphic evolution. Our results reveal that these surfaces are segmented by WNW-ESE striking faults, which initially experienced right-lateral transtensional movement followed by left-lateral strike-slip reactivation. Apatite fission-track (AFT) thermochronology of samples from relict surfaces yields AFT ages ranging from ~110 to ~100 Ma, while samples from fault zones record ages of 90-70 Ma. Thermal modeling of these samples indicates a period of moderate cooling in the mid-late Early Cretaceous, followed by a prolonged slow cooling phase for the relict surfaces. In contrast, fault zones show rapid cooling during the 90-70 Ma interval. By integrating these results with previous findings, we propose that the mid to late Early Cretaceous (~110-100 Ma) cooling event corresponds to extensional collapse following building of the Mongol-Okhotsk orogen. This process, coupled with increased humidity, enhanced erosion and relief reduction, facilitated the formation of low-relief surfaces. The influence of Mongol-Okhotsk orogenic collapse likely persisted into the Late Cretaceous (90–70 Ma), during which right-lateral transtensional faulting further segmented the landscape without generating significant topographic contrasts. By the Oligocene (~30 Ma), far-field effects from the India-Eurasia collision reactivated major faults in a left-lateral sense, driving regional uplift, surface tilting, and drainage incision. This uplift phase marked the end of landscape stability, as evidenced by increased sediment input into adjacent basins. Despite active faulting and fluvial incision, generally low erosion rates allowed for the preservation of large-scale Mesozoic low-relief surfaces.

## 1 Introduction

In orogenic belts worldwide, extensive high-elevation low-relief surfaces, typically forming summit plateaus or terraced pediments, bear testament to long-standing interactions between tectonic and surface processes (Phillips, 2002; Calvet et al., 2015; Whipple et al., 2017). Erosion rates drop considerably with decreasing topography, leading to the formation of low-relief surfaces, whose preservation is influenced by both tectonic stability and climatic conditions (e.g., van der Beek et al., 2009; Rohrmann et al., 2012; Morin et al., 2019). However, the processes behind their origin, preservation, and later modification remain a significant topic of debate. In particular, there is continued uncertainty regarding whether these low-relief surfaces developed at lower elevations and were subsequently uplifted during a phase of tectonic activity (Clark et al., 2006; Hetzel et al., 2011; Jaiswara et al., 2019) or whether they initially formed at high elevations within a tectonically uplifted environment (Liu-Zeng et al., 2008; van der Beek et al., 2009; Rohrmann et al., 2012; Cao et al., 2018). The distinction between these scenarios is challenging because it relies on differentiating surface-uplift from exhumation histories (e.g., England and Molnar,

1990). In both cases, the distribution of relict landscapes is primarily influenced by drainage reorganization, leading to localized erosion in captured basins and preservation of internally drained or inefficiently connected areas (Yang et al., 2015a; Jaiswara et al., 2019). Understanding these processes offers insight into long-term landscape evolution, the timing and role of tectonic activity, and how landscapes respond to large-scale climatic fluctuations, all of which are vital for comprehending the evolution of mountainous terrains over geological timescales.

Central Asia, and particularly regions such as the Tianshan, Gobi-Altai, and western Mongolia, hosts an extensive array of such low-relief surfaces that have attracted considerable scientific interest due to their potential to reveal complex histories of tectonic and climatic interactions (**Fig. 1**; Jolivet et al., 2007, 2010; Gillespie et al., 2017a; Morin et al., 2019). These Central Asian relict surfaces are thought to reflect periods of tectonic stability following major Mesozoic to early Cenozoic tectonic events, with phases of tectonic activity including the Triassic-Early Jurassic, Late Jurassic-Early Cretaceous, and Late

Cretaceous intervals (Jolivet et al., 2010, 2018; Gillespie et al., 2017a; Morin et al., 2019; He et al., 2021b). The tectonic history of the region is profoundly influenced by Mesozoic accretion-collision events along the southern Eurasian margin, commonly referred to as the Cimmerian collisions, as well as the Cenozoic India-Asia collision (Yin and Harrison, 2000; Kapp et al., 2007; Jolivet et al., 2010; Glorie and De Grave, 2016). These collisional events shaped the Tianshan region through cycles of uplift and erosion, leaving a lasting imprint on the landscape that includes the preservation of relict surfaces.

Additionally, studies have suggested that the Mesozoic orogenic and subsequent collapse phases of the Mongol-Okhotsk tectonic belt may correlate with exhumation episodes observed in the Tianshan Mountains (Gillespie et al., 2017a; Wang et al., 2018). The preservation of relict surfaces since the Mesozoic provides critical constraints on the long-term tectonic, climatic, and surface processes that have shaped the Tianshan region (Jolivet et al., 2010, 2018; Glorie et al., 2011; Gillespie et al., 2017a; Jolivet, 2017; He et al., 2021b).

In the easternmost part of the Tianshan, the Harlik Mountains (Harlik Shan or Karlik Tagh) stand out as a unique region where extensive high-elevation relict surfaces have been preserved along its southern flank. Thermochronological data suggest that these surfaces may have formed during the Late Cretaceous to early Paleogene, following a phase of rapid exhumation linked to tectonic activity (Gillespie et al., 2017a; Chen et al., 2020b). Subsequent limited erosion allowed for the long-term preservation and stabilization of these low-relief landscapes. Although it is generally agreed that late-Cenozoic reactivation of

bounding faults triggered the uplift of the Harlik Mountains, this phase of activity remains poorly constrained in terms of thermochronological data due to the minimal denudation associated with Cenozoic uplift (Cunningham et al., 2003; Gillespie et al., 2017a).

Although previous studies have explored the tectonic and climatic history of the eastern Tianshan, key questions remain regarding the formation, preservation, and modification of relict surfaces in the Harlik Mountains. Specifically, when and how

did these surfaces develop, what were the dominant tectonic and climatic controls on their evolution, and how have they been influenced by subsequent fault activity? To address these questions, this study integrates geomorphology, structural geology, and low-temperature thermochronology to reconstruct the long-term landscape evolution of the Harlik Mountains. We apply digital terrain analysis to delineate relict surfaces, assess their spatial distribution, and evaluate their relationship with regional faulting. Fluvial geomorphic analysis is conducted to examine channel steepness, knickpoint distribution, and drainage basin

morphology, providing insights into river incision history and landscape adjustment to tectonic forcing. Apatite fission-track (AFT) thermochronology is used to establish the thermal history of key relict surfaces and fault zones, constraining the timing of surface formation, exhumation, and fault reactivation. By combining these approaches, we aim to clarify the interplay between tectonic and climatic processes in shaping the Harlik Mountains and provide new insights into the broader landscape evolution of the eastern Tianshan.

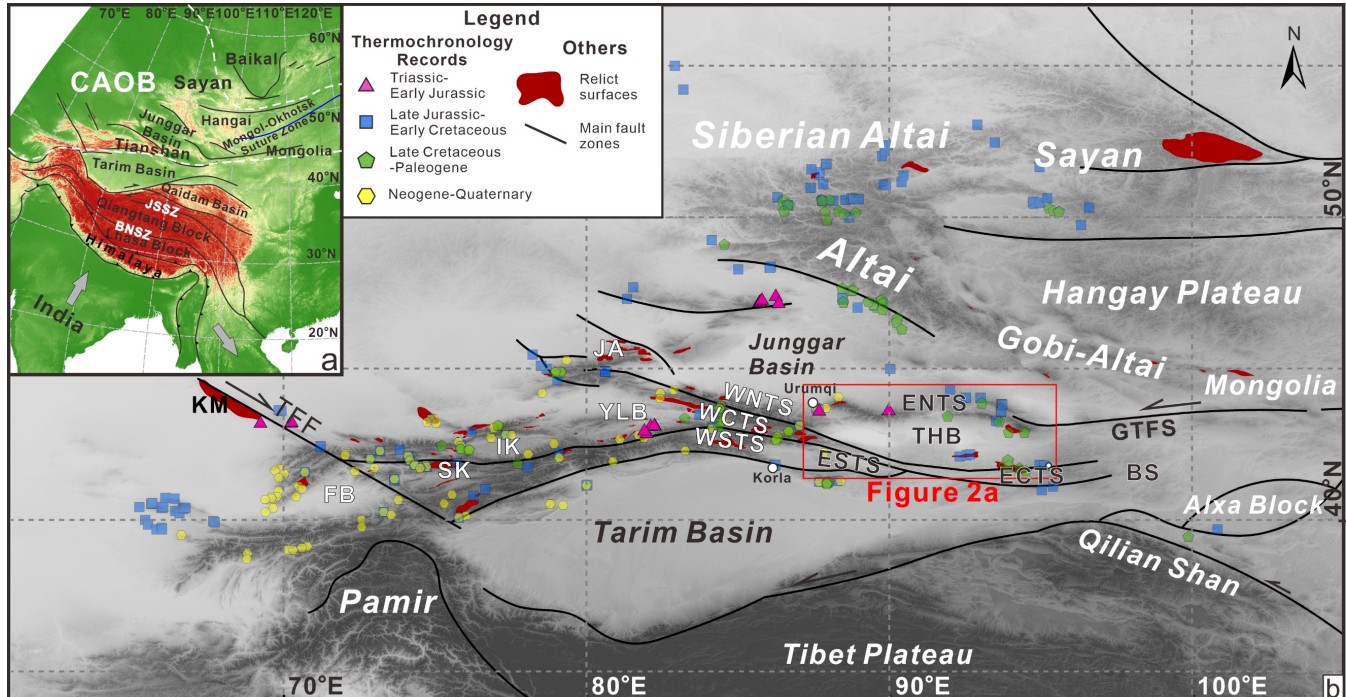


**Figure 1: a.** Tectonic sketch of the Central Asian Orogenic Belt (CAOB) and adjacent regions; **b.** Simplified topographic map of the Tianshan Orogenic Belt and adjacent areas, showing main tectonic units and faults. Rapid cooling episodes identified through thermochronology are marked, with data sourced from: De Grave and Van den haute (2002), Zhu et al. (2004, 2007), Shen et al. (2006), Sobel et al. (2006a, b), De Grave et al. (2007, 2008, 2014), Q. C. Wang et al. (2009), Zhang et al. (2009, 2011, 2016, 2021), 85 Glorie et al. (2010, 2011, 2012a, b, 2019, 2023), Jolivet et al. (2010), Lu et al. (2013), Macaulay et al. (2014), Gao et al. (2014), Bande et al. (2015, 2017), De Pelsmaeker et al. (2015), Jia et al. (2015), Sun et al. (2015, 2021), Tang et al. (2015), Kassner et al. (2016), Tian et al. (2016), Gillespie et al. (2017a, b, 2021), Chen et al. (2018, 2020a, b), Jepson et al. (2018a, b, c, 2021b), Nachtergaele et al. (2018), Song et al. (2018), Wang et al. (2018), Yin et al. (2018), He et al. (2021b, 2022a, 2022b, 2023; 2024), Zhimulev et al. (2021), Wu et al. (2023), and Jiang et al. (2024).

BNSZ: Bangong-Nujiang Suture Zone; BS: Beishan; ECTS: Eastern Central Tianshan Mountains; ENTS: Eastern North Tianshan Mountains; ESTS: Eastern South Tianshan Mountains; FB: Fergana Basin; GTFS: Gobi-Tianshan fault system; IK: Issyk-Kul; JA: Junggar Alatau; JSSZ: Jinsha Suture Zone; KM: Karatau Mountains; SK: Song-Kul; TFF: Talas–Fergana fault; THB: Turpan-Hami Basin; WCTS: Western Central Tianshan Mountains; WNTS: Western North Tianshan Mountains; WSTS: Western South Tianshan Mountains; YLB: Yili Block.

## 2 Background


### 2.1 Geologic setting

The Tianshan mountain range, located along the southern margin of the Central Asian Orogenic Belt (CAOB; **Fig. 1**), comprises three main tectonic units: the northern, central, and southern Tianshan (Windley et al., 2007; Charvet et al., 2011; Xiao et al., 2013). The Tianshan has a complex geological history, initially marked by the accretionary collision of several 100 microcontinents in the Paleozoic, followed by significant reworking due to strike-slip fault movement in the late Paleozoic (Windley et al., 1990, 2007; Xiao et al., 2013; He et al., 2021a). Subsequently, this region underwent several phases of intra-continental tectonic reactivation throughout the Mesozoic and Cenozoic (Windley et al., 1990; Dumitru et al., 2001). Notably, the southern Tianshan was subjected to significant compression and thrusting due to far-field stresses from the collision between the Indian and Eurasian plates, resulting in multiple large-scale tectonic deformation episodes (Bullen et al., 2001; 105 Sobel et al., 2006a; Macaulay et al., 2014; Bande et al., 2015; Kassner et al., 2016).

The Tianshan Orogenic Belt is geographically divided into western and eastern segments along the Urumqi-Korla line (**Fig. 1b**). These two regions exhibit distinct differences in geomorphic expression (**Fig. 1b**) as well as in their Mesozoic-Cenozoic tectonic evolution (Jolivet et al., 2010; Gillespie et al., 2017a; Chen et al., 2018; Sun et al., 2021), as discussed in **Section 2.2**. The western Tianshan is composed of the Western Chinese Tianshan and its extension into the Tianshan ranges of Kazakhstan, Kyrgyzstan, and Uzbekistan (**Fig. 1**). The eastern Tianshan, which encircles the Turpan-Hami Basin, includes the Moqinwula, Bogda-Barkol-Harlik, and Jueluotage ranges from north to south (**Figs. 1 and 2**). The easternmost Tianshan forms part of the Harlik arc, a region that developed in response to subduction of the Paleo-Asian Ocean and consists primarily of Ordovician-Carboniferous volcanic and volcaniclastic rocks (Xiao et al., 2004; Deng et al., 2016; Ji et al., 2019; Ni et al., 2021). In addition, Carboniferous and Permian igneous rocks are widespread in the easternmost Tianshan (Ma et al., 2015). The Harlik Mountains are also dominated by Paleozoic strata, with its southern flank hosting sizable granitoid and diorite plutons, predominantly of Silurian and Carboniferous age (**Fig. 2c**; Ma et al., 2015; Gillespie et al., 2017a).

The Harlik Mountains lie at the boundary between the northern Tianshan and the East Junggar arc or terrane, a location that has experienced a multi-phase tectonic evolution during the Mesozoic and Cenozoic (Chen et al., 2020a). The Harlik Mountains are structurally linked to the Gobi-Tianshan fault system, a significant left-lateral strike-slip fault system that connects the eastern Tianshan with the Gobi-Altai. This fault system forms a restraining-bend termination zone featuring a horsetail fault geometry (**Fig. 1**; Cunningham et al., 2003; Cunningham, 2013). The Harlik Mountains thus form an asymmetrical fault-bounded structure, delimited by the North Harlik Boundary Fault (NHBF), South Harlik Boundary Fault (SHBF), and Barkol Fault System (BFS) (**Fig. 2c**). This structure is marked by substantial uplift and denudation of the Paleozoic basement rocks to the north, while the southern flank is capped by a gently dipping relict surface (Gillespie et al., 2017a).

Since the Mesozoic, tectonic activity in the easternmost Tianshan has been influenced by a range of tectonic settings, resulting in a complex structural system. During the Late Jurassic to Early Cretaceous, deformation transitioned from compression to extension, evidenced by Late Jurassic-Early Cretaceous folding and thrust faulting in the Moqinwula Mountains (Chen et al., 2020a) and mid- to late Early Cretaceous normal faulting in the Barkol Mountains (Chen et al., 2020b). Cenozoic tectonic activity in the Harlik Mountains is generally associated with the far-field effects of the India-Asia collision, which reactivated pre-existing fault systems and generated a characteristic southward-tilted structure (Cunningham et al., 2003; Gillespie et al., 2017a). The structural framework of the Harlik Mountains also reflects a regional transition in deformation style, from thrust-dominated deformation in the central Tianshan to transpressional deformation in the easternmost Tianshan, driven by the interaction between the rigid Tarim craton and the more deformable Junggar block (Cunningham et al., 2003; Cunningham and Zhang, 2021).

**2.2 Mesozoic-Cenozoic evolution of relict surfaces in the Tianshan Mountains**

In response to the Cenozoic India-Asia collision, individual ranges within the Tianshan were uplifted mainly along pre-existing faults, resulting in a basin-and-range type landscape (Sobel et al., 2006a; Jolivet et al., 2010; Chen et al., 2018; Chang et al., 2019). Despite significant uplift, extensive relicts of the low-relief Mesozoic to early Cenozoic topography are widely distributed throughout the Tianshan region (Jolivet et al., 2010, 2018; Morin et al., 2019; He et al., 2023). These relict surfaces provide evidence of multiple phases of erosion, tectonic stability, and subsequent reactivation. However, the processes underlying the development of these relict low-relief surfaces vary across different regions of the Tianshan.

**2.2.1 Kazakh and Kyrgyz Tianshan**

In western Kazakhstan, southwest of the Talas-Fergana fault (**Fig. 1**), Allen et al. (2001) identified a tilted relict surface atop the Karatau Mountains. Although this surface has been dissected during the late Cenozoic, the age of its formation remains undetermined. In Kyrgyzstan, low-temperature thermochronology data indicate that high-altitude relict surfaces in the Issyk-

Kul and Song-Kul areas were formed during a prolonged period of slow erosion, following a phase of significant exhumation in the Late Triassic to Early Jurassic (De Grave et al., 2011, 2013; Rolland et al., 2020). A sedimentary hiatus spanning the Middle Jurassic to Paleogene in surrounding basins, coupled with Neogene-Quaternary sediments partially covering these surfaces, suggests that they developed between the Jurassic and Paleogene (Burbank et al., 1999; De Pelsmaeker et al., 2018). However, limited apatite fission-track thermochronology data imply that the Kyrgyz Tianshan experienced rapid exhumation during the Cretaceous (Sobel et al., 2006a; De Grave et al., 2012; De Pelsmaeker et al., 2015). Chen et al. (2018) proposed that the low-relief surfaces in Kyrgyzstan emerged during a phase of tectonic stability between the Late Cretaceous and early Cenozoic.

### 2.2.2 Chinese western Tianshan

The tectonic and geomorphic evolution of the Chinese western Tianshan during the Meso-Cenozoic parallels that of the Kyrgyz Tianshan. In regions such as the Yili block, Western Central Tianshan, and Western South Tianshan, low-temperature thermochronology data indicate significant exhumation during the Late Triassic to Early Jurassic, followed by slower exhumation from the Middle Jurassic to Late Cretaceous (Jolivet et al., 2010; Wang et al., 2018; Glorie et al., 2019; He et al., 2021b, 2022b). The degree to which Late Jurassic to Early Cretaceous exhumation affected the Chinese western Tianshan remains a topic of discussion (Dumitru et al., 2001; Jolivet et al., 2010; Wang et al., 2018; Glorie et al., 2019; Xiang et al., 2019). According to Jolivet et al. (2010), Wang et al. (2018), and He et al. (2021b), the Chinese western Tianshan did not experience significant relief building during its Mesozoic evolution; instead, the low-relief surfaces were established during this period. A geomorphological and sedimentological investigation in the northern Chinese western Tianshan (Jolivet et al., 2018) demonstrated that the Mesozoic relict surfaces transformed into multi-phased, nested erosional surfaces, which were subsequently incised by Late Cretaceous to early Paleogene paleo-valleys.

### 2.2.3 Eastern Tianshan

Recent studies have increasingly focused on the eastern Tianshan, where relict surfaces and significant tectonic structures have been identified. Thermal-history modeling of thermochronology data indicates that the primary exhumation phase in the Jueluotage Range occurred during the Triassic to Early Jurassic (Gong et al., 2021; Sun et al., 2021). This was followed by multiple exhumation pulses recorded during the Early Cretaceous and Late Cretaceous-Paleogene (Gao et al., 2014; Sun et al., 2015, 2021; Gong et al., 2021; He et al., 2022a). Different datasets independently suggest a decrease in relief in the Jueluotage range during the Jurassic. Detrital zircon geochronology from the Turpan-Hami Basin reveals a significant decline in provenance linked to the Jueluotage Range during the Middle to Late Jurassic (Shen et al., 2020; Qin et al., 2022). Both the Turpan-Hami Basin and parts of the Jueluotage Range experienced burial during Jurassic deposition (Shao et al., 1999, 2003; Gong et al., 2021; Sun et al., 2021). The low relief and elevation of the Jueluotage Range are presumably due to prolonged slow erosion during the Mesozoic and Cenozoic.

Previous thermochronological studies of the Bogda Mountains have primarily concentrated on the western portion near Urumqi. AFT ages in the Bogda Mountains display a younging trend from southwest to northeast, reflecting progressive uplift and exhumation from the Late Jurassic to the Miocene (Tang et al., 2015). Thermal-history modeling has identified two rapid exhumation events: one during the latest Jurassic to Early Cretaceous and another during the Oligocene to Miocene (Tang et al., 2015). An erosional surface (**Fig. 2a**) that truncates Paleozoic rocks in the Bogda Mountains is thought to have formed during a Mesozoic to Paleogene period of slow erosion between these two rapid exhumation events (Tang et al., 2015; Morin et al., 2019; He et al., 2023).

In the easternmost Tianshan, most thermochronological data indicate exhumation primarily during the Early to Middle Triassic, Early Cretaceous, and Late Cretaceous to Paleocene (Gillespie et al., 2017a; Chen et al., 2020a, 2020b; He et al., 2022a). Cunningham et al. (2003) documented a tilted low-relief surface preserved in the Harlik Mountains (**Fig. 2a**). Gillespie et al.

(2017a) reported evidence of Late Cretaceous exhumation through thermal-history modeling of a sample collected from the edge of this surface, suggesting that the surface formed during the Late Cretaceous to early Paleogene and was subsequently
tilted during the Cenozoic. Structural analysis and low-temperature thermochronology data indicate that the Moqinwula Mountains underwent two stages of deformation during the Late Jurassic and Late Cretaceous (Chen et al., 2020a). Chen et al. (2020a) proposed that the low-relief surfaces preserved on the summits of the Moqinwula Mountains were formed in the Early Cretaceous, between these two deformation stages.

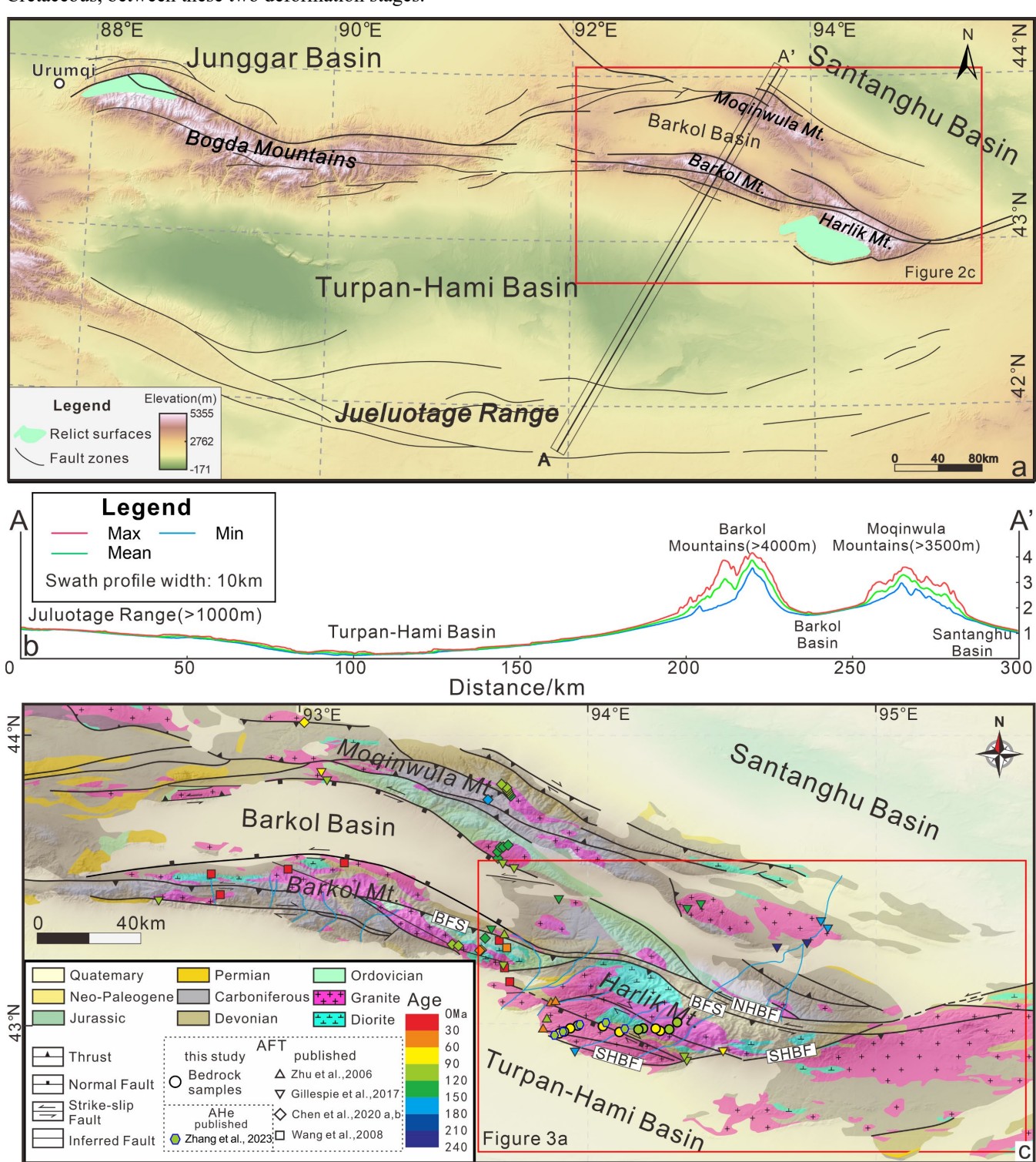

**Figure 2: a. Map showing the topography of the eastern Tianshan and its main tectonic structures; b. topographic cross-section A-A' from the Santanghu Basin to the Jueluotage Range; c. Simplified geological map of the Easternmost Tianshan (modified from China Geological Survey, 2007)**

## 3 Tectonic Geomorphology of the Harlik Mountains

To investigate the geomorphological features of the Harlik Mountains and their relationships with the structure of the mountain range, we integrated Digital Elevation Model (DEM)-based terrain analysis, fluvial geomorphic analysis, and structural analysis. DEM-based analysis allows identification of relict surfaces, establishing their spatial distribution and morphological characteristics. Fluvial geomorphic analysis evaluates river incision and drainage evolution, revealing the interplay between surface evolution and fluvial erosion. Structural analysis constrains fault kinematics and paleo-stress regimes, clarifying tectonic controls on landscape evolution. This integrated approach enables a comprehensive reconstruction of the deformation history of the range and its geomorphic impacts.

### 3.1 DEM-based terrain analysis

Although relict surfaces in the Harlik Mountains have been recognized for nearly two decades, studies have primarily focused on their mapping and delineation using remote-sensing imagery and field surveys (Cunningham et al., 2003; Gillespie et al., 2017a). Morphologically, these surfaces are typically manifested as elevated positive relief landforms characterized by sub-horizontal or slightly tilted planar topography (Calvet et al., 2015; Yang et al., 2015a; Liu et al., 2019).

#### 3.1.1 Materials and Methods

We employed digital terrain analysis based on the Shuttle Radar Topography Mission (SRTM) DEM, which has a resolution of 3 arc seconds (~90 m), to extract relict surfaces in the Harlik Mountains. Low-relief areas within and around the Harlik Mountains were delineated using a topographic slope criterion of less than 14° (Calvet et al., 2015; Haider et al., 2015; Liu et al., 2019). These areas encompass potential relict surfaces as well as low-relief regions in valleys and basins. To isolate the relict surfaces within the mountain range, we excluded areas with a relative height of less than 40 m (**Fig. 3a**). The relative height is defined as the elevation difference between the original DEM and a reference erosion surface (Haider et al., 2015). We extracted the drainage network from the DEM to identify river systems that have incised the original low-relief surfaces, using a threshold contributing area of 100 cells. A reference erosion surface was created by interpolating the elevations corresponding to the valley bottoms (i.e., the drainage lines). **Figure S1** provides a detailed description of the extraction procedure, along with the original DEM and slope map.

#### 3.1.2 Results

Based on the degree of preservation and the orientation of individual low-relief areas, we categorized the relict surfaces into several blocks (S1 to S9), as shown in **Fig. 3a**. To better understand the spatial relationships among the different relict surfaces, we extracted several NE-SW and NW-SE trending swath profiles (**Fig. 3b-f**). The NW-SE trending swath profiles reveal that elevations of the relict surfaces on the southern flank of the Harlik Mountains increase towards the east. Additionally, the relict surfaces in the eastern part of the range exhibit more pronounced erosion compared to those in the west (**Fig. 3b-d**). The NE-SW trending swath profiles illustrate significant topographic contrasts among these relict surfaces, with varying slopes dipping towards the south or southwest (**Fig. 3e-f**). Field observations indicate that these topographic contrasts are controlled by faults, which are discussed further in Section 3.2.

We extracted slope and aspect data for the grid points corresponding to the relict surfaces in different blocks, presented as slope-aspect rose diagrams (see **Figure S2**). The primary aspects of each surface are indicated with arrows in **Fig. 3a**. Some surfaces are represented by fan-shaped symbols, reflecting an angular range of aspects due to the absence of distinct individual aspects. The orientations of these surfaces are significantly influenced by boundary faults, including the North Harlik Boundary Fault (NHBF), the South Harlik Boundary Fault (SHBF), and the Barkol Fault System (BFS). Surface S1, bounded by the NHBF and BFS, features a gentle slope with a predominant westward aspect. Surface S3 is also bounded by the NHBF and BFS, gently dipping towards the northeast. Surface S2, located north of the NHBF, dips northeastward, perpendicular to the

orientation of the NHBF. To the south of the BFS, surfaces S4, S5, and S7 predominantly dip towards the southwest. These surfaces are constrained to the north by the BFS and appear to be cut and variably tilted by different strands of the SHBF.

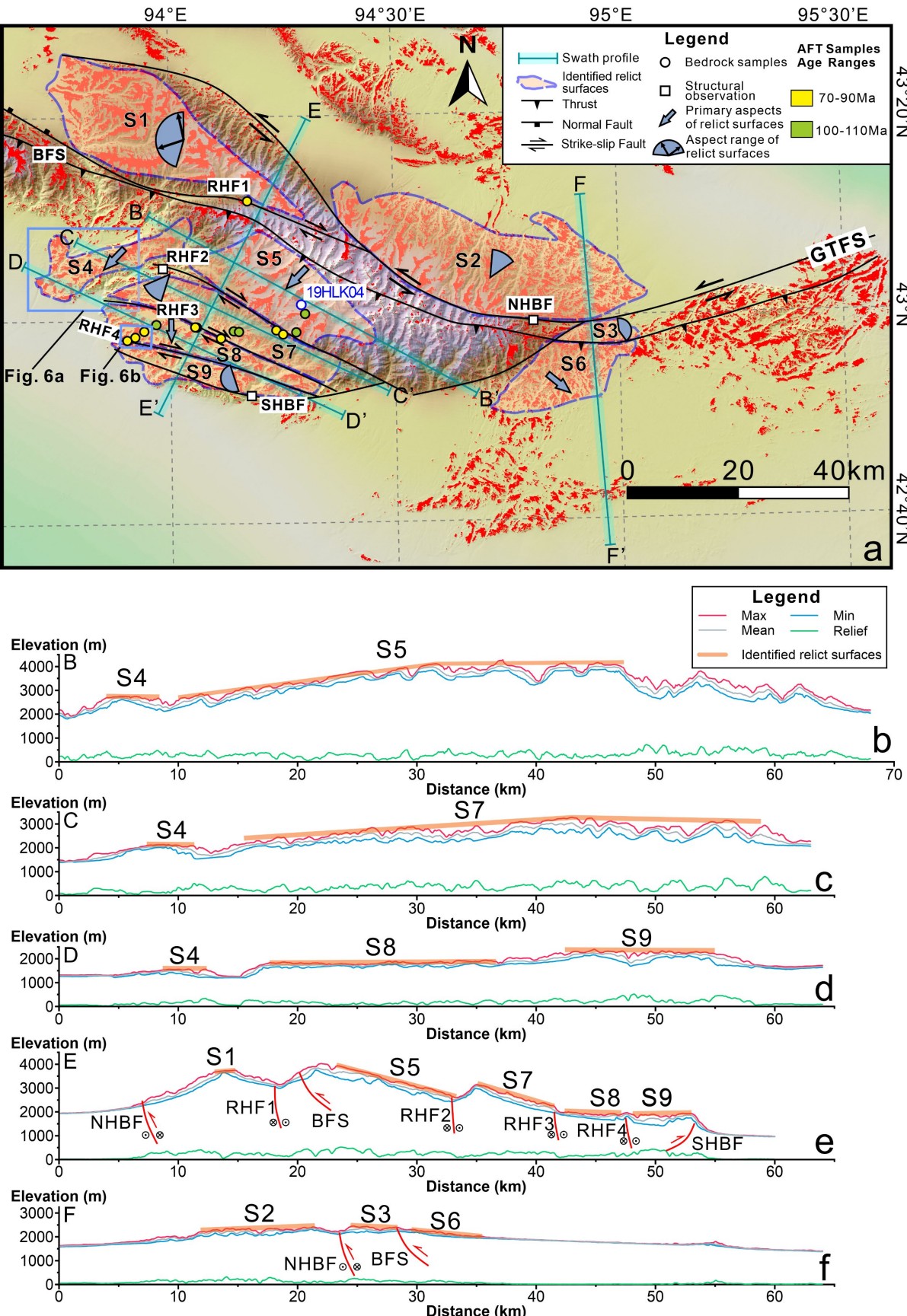

Figure 3: a. DEM of the Harlik Mountains with low-relief areas (slope <14°) in red; different identified relict surfaces (indicated as S1 – S9) are shown with pink shading and dashed outline; b-f. topographic cross-sections B-B' to F-F' (see location in panel a). AFT: Apatite fission-track; BFS: Barkol Fault System; NHBF: North Harlik Boundary Fault; RHF: Right-lateral strike slip fault; SHBF: South Harlik Boundary Fault.

## 3.2 Fluvial Geomorphic Analysis

Fluvial erosion has modified the relict low-relief surfaces on the southern flank of the Harlik Mountains, reflecting the interplay between river incision and tectonic deformation. Using the 3-arc-second resolution SRTM DEM, we analyzed drainage metrics ($k_{sn}$ and $\chi$) to quantify incision patterns, detect transient landscape adjustments, and assess structural controls on river evolution.

### 3.2.1 Methods

Catchments and fluvial metrics, such as the normalized steepness index ($k_{sn}$) and the integral proxy ($\chi$), were extracted using the Topographic Analysis Kit (Forte and Whipple, 2019) alongside TopoToolbox 2 algorithms (Schwanghart and Scherler, 2014). Ten basins draining the southern flank of the Harlik Mountains were extracted based on a minimum drainage area threshold of $10^6$ m². The slope ($S$) and contributing drainage area ($A$) of the river profile follow a power-law relationship (e.g., (Flint, 1974; Whipple and Tucker, 1999):

$$S = k_s \, A^{-\theta} \tag{1}$$

The steepness index ($k_s$) allows comparing channel gradients across different drainage areas (Flint, 1974; Whipple and Tucker, 1999; Wobus et al., 2006; Kirby et al., 2012), while the concavity index ($\theta$) quantifies the downstream decline in river gradient (Lague, 2014; Gailleton et al., 2021; Smith et al., 2022). Chi ($\chi$) analysis is a robust method for assessing river profile evolution and detecting transient landscape adjustments. The $\chi$ parameter is defined as:

$$\chi = \int_{x_b}^{x} \left( \frac{A_0}{A(x')} \right)^{\frac{m}{n}} dx' \tag{2}$$

where $x_b$ represents the base-level position of the river (e.g., the river mouth), $A(x')$ denotes the upstream catchment area, $m$ and $n$ are empirical constants, and $A_0$ is an arbitrary scaling area. Under steady state, the ratio $m/n$ corresponds to $\theta$.

We determined an optimal concavity index, $\theta_{ref}$, for the southern Harlik catchments using Bayesian optimization (**Fig. S3**; Schwanghart and Scherler, 2017). The optimized $\theta_{ref}$ (~0.22) was then used to calculate the normalised steepness index, $k_{sn}$, for the river channels across all ten basins. We selected three basins (1, 3, and 4) for further longitudinal profile and $\chi$ analysis, with $A_0$ set to $10^6$ m².

We set a 120 m tolerance for knickpoint extraction in longitudinal profiles using TopoToolbox 2 (Schwanghart and Scherler, 2014). To minimize misidentifications from DEM noise, we validated the extracted knickpoints using satellite imagery and $\chi$–z plots. Knickpoints were then classified based on their likely origins as (1) glacial, (2) structural (potentially linked to active faults), or (3) transient, resulting from drainage reorganization or changes in tectonic uplift rates (Marrucci et al., 2018; Gong et al., 2024).

### 3.2.2 Results

The distribution of channel steepness ($k_{sn}$) aligns with the overall topographic pattern. High $k_{sn}$ values are primarily concentrated along faults, in the eastern part of the area where no relict surfaces are preserved, and, to a lesser extent, on Surface S7, whereas low values are observed on Surfaces S5, S8, and S9 (**Fig. 4a**). Additionally, $k_{sn}$ exhibits an increasing trend toward the east across all relict surfaces on the southern flank of the Harlik Mountains. A total of 16 knickpoints were extracted from the three selected basins (**Fig. 4a**). Glacial knickpoints are mainly found at the base of U-shaped valleys, often occurring at tributary junctions or within hanging valleys, with elevations ranging from 3,000 to 3,500 m. In longitudinal profiles, they appear as distinct concave-up breaks (**Figs. 4b-d**), while in $\chi$–z plots, they exhibit a pronounced deviation from linearity (**Figs. 4e-g**). Structural knickpoints in this study are exclusively located near faults, typically within valleys incising relict surfaces. They are manifested as discrete heterogeneities along longitudinal profiles (**Figs. 4c-d**), indicating localized variations in channel gradient. Transient knickpoints are concentrated along tributaries extending to relict surfaces (e.g., S7–S9) and are

consistently positioned near these surfaces. In longitudinal profiles, they are marked by a notable downstream
increase in slope (**Figs. 4b-d**), accompanied by a similar trend in $k_{sn}$ (**Fig. 4a**). In χ–z plots, they appear as prominent
convex breaks along the curve (**Figs. 4e-g**).

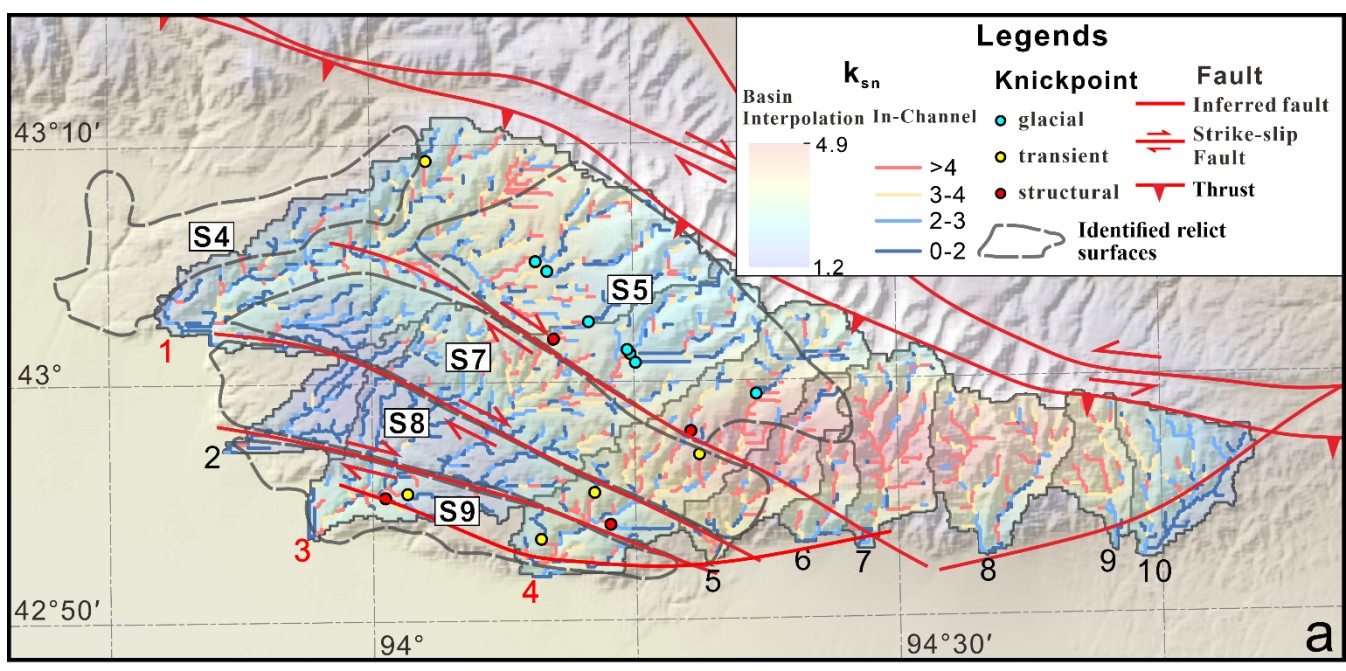

## Longitudinal Profile:

## χ –z plot:

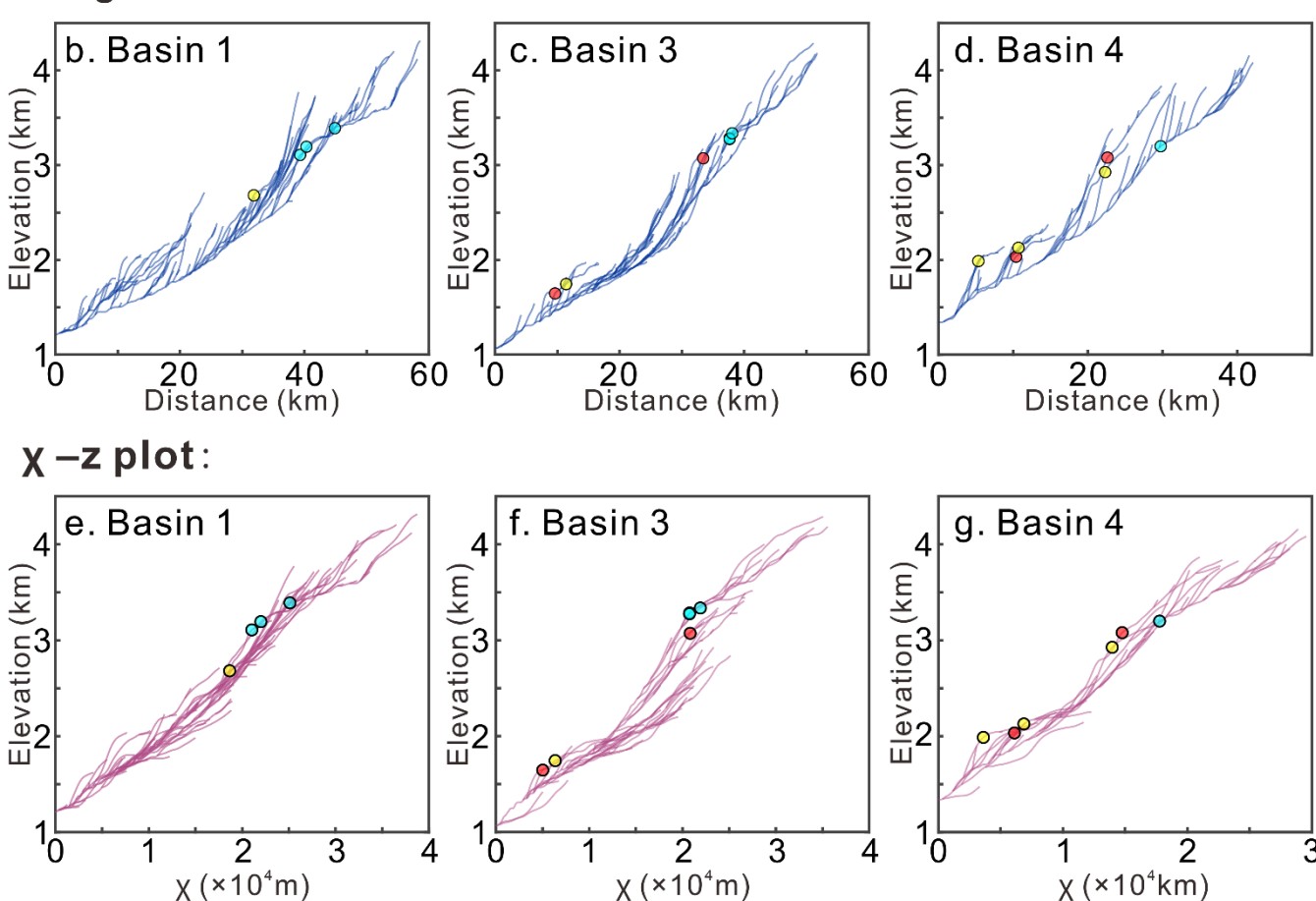

**Figure 4: a. Normalized channel steepness ($k_{sn}$) map of catchments draining the southern flank of the Harlik Mountains and locations**
**of knickpoints with inferred origins overlain on shaded topographic relief. b-d. Longitudinal profiles of the main trunk and**

tributaries for catchments 1, 3, and 4, with the extracted knickpoints marked; e-g. Corresponding χ plots of these catchments, with the knickpoints indicated at their corresponding positions in χ-space.

## 3.3 Structural analysis

The relict surfaces in the Harlik Mountains are bounded by prominent topographic scarps visible on swath profiles, which
correspond to linear boundaries in map view (**Fig. 3**) that correspond to faults. To better understand the formation and tectonic evolution of these surfaces, we conducted a detailed structural analysis of the Harlik fault system through field investigations.

### 3.3.1 Methods

In the field, we conducted a systematic investigation of the fault system in the Harlik Mountains. Fault kinematics were identified based on structural features such as slickenlines, steps, en-echelon cracks, and offset markers, including veins and
300 volcanic tuff bands. We measured the attitude of fault planes and slickenlines and performed paleostress analysis using ***Win-Tensor*** software (Delvaux and Sperner, 2003; **Fig. 5**). Win-Tensor inverts fault-plane attitudes and slickenline orientations, using iterative optimization to refine principal stress axes and the stress ratio for accurate stress field reconstruction. To supplement the field observations, Google Earth imagery was utilized to identify fault-controlled landforms and accurately map fault traces (**Fig. 6**).

### 3.3.2 Results

The east-west striking NHBF delineates the northern edge of the Harlik Mountains. This fault exhibits a southward bend before extending eastward, where it merges with the Gobi-Tianshan fault (**Fig. 3a**). Field observations reveal that the NHBF displays predominantly left-lateral oblique strike-slip movement, with the southern block moving upward relative to the northern block. Slickenlines on the fault surface suggest a principal compressional stress direction oriented NE-SW (**Fig. 5m**). Similar to the
310 NHBF, the SHBF also features low-angle thrust faults combined with high-angle left-lateral strike-slip faults (**Figs. 5b-c**). Surface S1 is bordered by the NHBF to the north and by a right-lateral, NW-SE trending strike-slip fault (RHF 1) to the south (**Figs. 3 and 4d**). Additionally, three parallel NW-SE trending faults are present along the southern flank of the range, each demonstrating right-lateral strike-slip motion (**Figs. 5e-g**). These faults have fragmented the relict surfaces on the southern flank into separate blocks, each dipping southwestward at various angles. Paleo-stress analysis of RHF 2 and RHF 3 indicates
NE-SW oriented extension, corresponding to the right-lateral movement observed along these faults (**Figs. 5m**). Field evidence also shows that ENE-WSW trending left-lateral strike-slip faults subsequently overprinted these NW-SE right-lateral faults. In samples collected from these strike-slip faults, hydrothermal features are evident, including veins of siliceous and carbonate fluids associated with fault activity, along with chloritization and epidotization alterations. These characteristics differ markedly from samples taken from the low-relief surfaces, suggesting localized fluid-rock interaction during fault activity
(**Figs. 5j-l**).

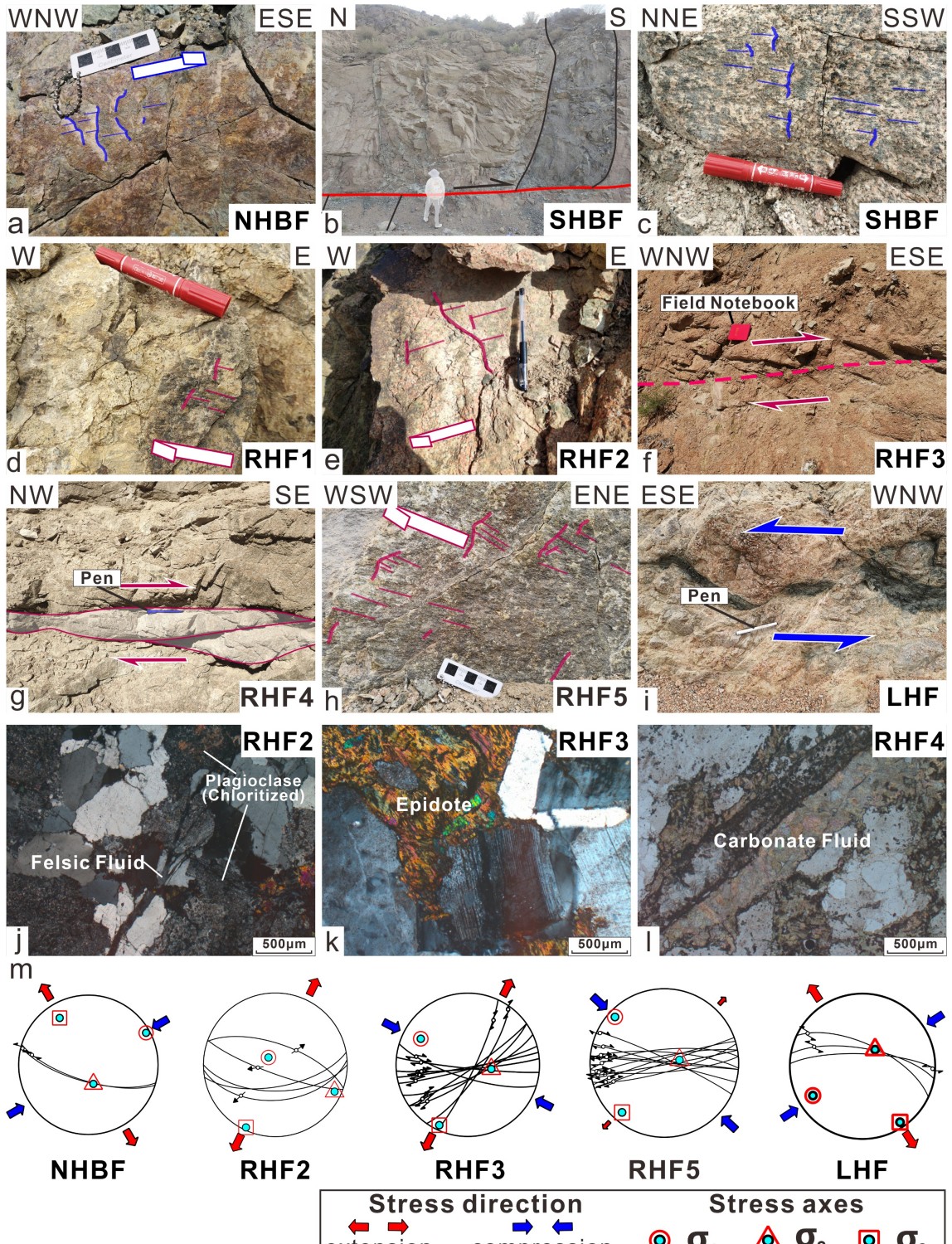

**Figure 5: Outcrop photos of faults in the Harlik Mountains (a-i; locations of the study sites are shown in Fig. 3a; High-resolution versions are provided in Figure S4) and microstructural images of rocks within the fault zones (j-l). a. Slickenlines and steps on the NHBF fault surface indicate left-lateral oblique thrust motion. b. A low-angle thrust fault delineating the southern boundary of the Harlik Mountains, with movement indicated by offset veins. c. Slickenlines and steps on the SHBF fault surface, reflecting left-lateral movement. d. Slickenlines and steps on the RHF 1 fault surface, indicative of right-lateral movement. e. Slickenlines and steps on the RHF 2 fault surface, also indicating right-lateral movement. f. En-echelon cracks on RHF 3 (viewed from above), demonstrating right-lateral motion. g. Asymmetric granitic lenses (top view) indicative of dextral shear along the RHF 4. h. Slickenlines and steps on the RHF 5 fault surface, showing dextral movement. i. The offset of volcanic tuff bands (top view) indicates sinistral (left-lateral) movement. j. Chloritization of plagioclase and siliceous fluid vein observed in the RHF 2. k. Epidote alteration in RHF 3. l. Carbonate fluid vein in RHF 4; l. paleo-stress orientations of NHBF, RHF2, RHF3, RHF5, and LHF. Detailed fault data are provided in Table S1.**

Field mapping and geomorphological analysis in the Shichengzi area of the southwestern Harlik Mountains (**Fig. 3a**) reveal a NW-SE trending right-lateral strike-slip fault (RHF 5; **Fig. 5h**) and an ENE-WSW trending left-lateral strike-slip fault (LHF; **Fig. 5i**). Remote-sensing imagery reveals that the LHF intersects RHF 5 (**Fig. 6a**), providing relative age constraints for these fault systems. Inferred stress tensors for these fault sets indicate NW-SE compression for RHF 5 (**Fig. 5m**) and NE-SW compression for LHF (**Fig. 5m**), further elucidating the complex, multi-phase deformation history of the Harlik fault system.

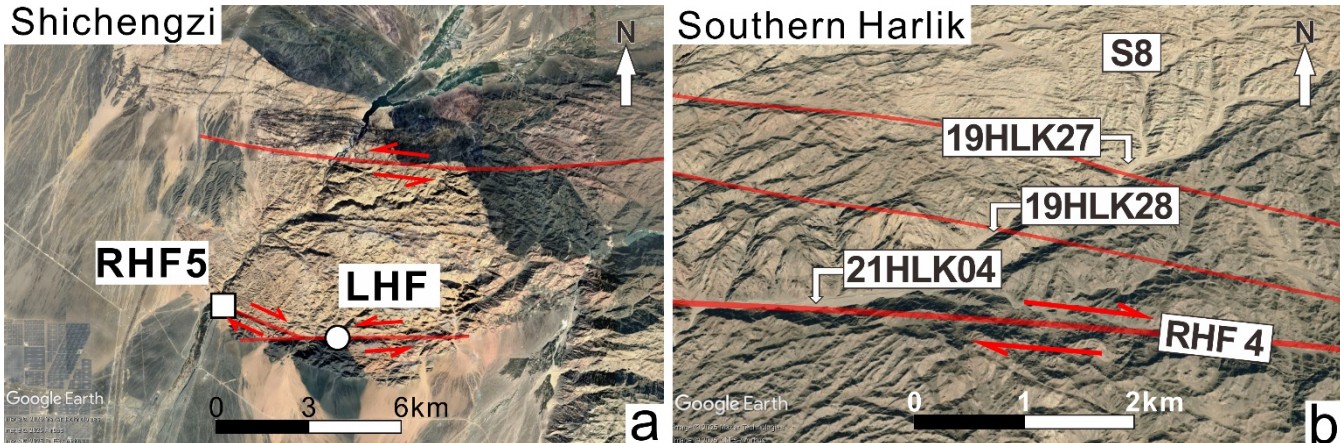

**Figure 6: Google Earth imagery of the Shichengzi area (a) and RHF 4 (b); see Fig. 3a for locations. Background image: © Google Earth, Image © 2025 CNES / Airbus, Airbus, Maxar Technologies.**

## 4 Low-temperature thermochronology

Low-temperature thermochronology was employed to elucidate the evolutionary history of low-relief surfaces in the Harlik Mountains. This technique serves as a valuable tool for constraining both the magnitude and timing of denudation, as fluctuations in denudation rates are often correlated with the formation, preservation, and degradation of low-relief surfaces in mountainous regions (e.g., van der Beek et al., 2009; Rohrmann et al., 2012; Morin et al., 2019; Cao et al., 2022). Typically, tectonic activity can cause rapid exhumation, leading to relief development and fast rock cooling. As tectonic activity decreases, erosion lowers the relief, resulting in a period of slow rock cooling linked to slow denudation. A phase of thermal stability, with no significant vertical movements, corresponds to low-relief topography, until tectonic reactivation leads to renewed uplift and exhumation. In the Harlik Mountains, relict surfaces are fragmented into several blocks by faults and incised valleys, indicating a complex history of relief rejuvenation. Thus, rocks situated on these relict surfaces are expected to preserve cooling histories reflective of both relief-building and subsequent leveling processes, while those located near faults or within valleys may record late-stage cooling events linked to relief rejuvenation (e.g., Cao et al., 2022).

### 4.1 Sampling strategy

While previous studies have reported low-temperature thermochronology data from the Harlik Mountains, the majority of these data have been collected from the margins of the mountain belt or from the main valleys within the mountains (Zhu et al., 2006; Gillespie et al., 2017a; He et al., 2022a; Zhang et al., 2023). Consequently, the relationship between these samples and the relict surfaces or fault systems remains ambiguous. To investigate the relief evolution in the Harlik Mountains, we systematically collected samples for low-temperature thermochronology along two valleys situated on the southern flank of the mountain range. The samples encompass the four main identified blocks and the fault zones separating them. Sampling was conducted at an average elevation interval of 200-300 m, ranging from 1155 m to 3747 m (**Table 1**). A detailed description of the samples, including their geographic locations and AFT age data, is provided in **Table 1**.

**Table 1 Sample information and apatite fission track data, southern Harlik Mountains.**

| Location | Sample Information | | | | | | ρs (10⁵ cm⁻²) | Ns | ²³⁸U (ppm) | Ages | | | | Track lengths | | | Mean D$_{par}$ (μm) |
| | No. | Location | | | Lith. | n | | | | Pooled age (Ma± 1σ) | P(χ²) (%) | Dispersion (%) | Central age (Ma± 1σ) | MTL (μm) | SD (μm) | N | |
| | | Lon. | Lat. | Elev. (m) | | | | | | | | | | | | | |
|---|---|---|---|---|---|---|---|---|---|---|---|---|---|---|---|---|---|
| RHF 1 fault zone | 23HLK02 | 94.175 | 43.194 | 2783 | Granite | 25 | 6.87 | 553 | 15.63 | 90±5 | 9.1 | 14 | 91±5 | 12.9 | 1.2 | 100 | 2.5 |
| S5 | 19HLK01 | 94.280 | 42.979 | 3390 | Granite | 15 | 4.33 | 156 | 8.03 | 109±5 | 100 | 0 | 110±8 | 12.5 | 2.1 | 35 | 1.1 370 |
| | 19HLK02 | 94.300 | 43.009 | 3747 | Granite | 16 | 6.98 | 454 | 12.93 | 102±3 | 100 | 0 | 102±6 | 12.4 | 1.5 | 105 | 1.2 |
| RHF 2 fault zone | 19HLK07 | 94.249 | 42.974 | 3097 | Diorite | 18 | 18.46 | 660 | 52.98 | 70±5 | 2.8 | 15 | 73±4 | 12.1 | 1.2 | 140 | 1.5 |
| | 19HLK08 | 94.235 | 42.982 | 2799 | Diorite | 18 | 12.21 | 643 | 29.38 | 84±5 | 33 | 7 | 86±5 | 12.4 | 1.2 | 82 | 1.5 |
| | 19HLK10 | 94.197 | 42.986 | 2474 | Granite | 20 | 5.71 | 336 | 13.43 | 84±5 | 90 | 15 | 86±6 | 12.6 | 1.2 | 61 | 1.2 |
| S7 | 19HLK14 | 94.152 | 42.981 | 2152 | Granite | 17 | 5.37 | 369 | 9.97 | 110±3 | 100 | 0 | 111±7 | 12.8 | 1.3 | 58 | 1.4 |
| | 19HLK15 | 94.141 | 42.982 | 2026 | Granite | 18 | 6.06 | 335 | 10.96 | 106±4 | 97 | 17 | 107±7 | 12.5 | 1.6 | 26 | 1.4 375 |
| RHF 3 fault zone | 19HLK17 | 94.112 | 42.970 | 1849 | Diorite | 15 | 5.78 | 295 | 14.82 | 77±5 | 96 | 0 | 78±6 | 12.1 | 1.2 | 35 | 1.4 |
| | 19HLK22 | 94.058 | 42.989 | 1847 | Granite | 19 | 6.35 | 260 | 18.96 | 80±5 | 55 | 0 | 80±5 | 12.4 | 1.2 | 28 | 1.4 |
| | 19HLK24 | 94.055 | 42.989 | 1869 | Granite | 17 | 8.75 | 236 | 20.69 | 74±4 | 91 | 0 | 76±6 | 12.9 | 0.9 | 18 | 1.5 |
| S8 | 19HLK26 | 93.968 | 42.993 | 1619 | Granite | 15 | 3.66 | 201 | 8.92 | 102±3 | 100 | 0 | 103±8 | 12.0 | 1.4 | 43 | 1.2 |
| | 19HLK27 | 93.941 | 42.983 | 1516 | Granite | 13 | 8.42 | 276 | 21.57 | 70±6 | 75 | 19 | 72±6 | 12.6 | 1.2 | 50 | 1.2 |
| | 19HLK28 | 93.921 | 42.973 | 1398 | Granite | 17 | 3.2 | 89 | 8.70 | 84±7 | 97 | 0 | 87±9 | 12.4 | 1.4 | 31 | 1.1 380 |
| RHF 4 | 21HLK04 | 93.903 | 42.968 | 1317 | Granite | 23 | 9.19 | 385 | 26.39 | 70±5 | 17 | 10 | 73±4 | - | - | - | - |

Note: Uranium content of all grains measured by LA-ICP-MS. Pooled AFT ages of all grains. Central age calculated using the RadialPlotter program (Vermeesch, 2009).
Column headings are as follows: **No.:** Sample identification number. **Lon., Lat.:** Longitude and latitude of the sampling location. **Elev. (m):** Elevation in meters above sea level. **Lith.:** Lithology of the sample. **n:** Number of analyzed grains. **ρs (10⁵ cm⁻²):** Surface density of spontaneous fission tracks. **Ns:** Number of spontaneous fission tracks counted. **²³⁸U (ppm):** Uranium concentration in parts per million. **Pooled age (Ma ± 1σ):** Pooled fission track age in million years with one standard deviation. **P(χ²) (%):** Chi-square probability that the single-grain ages represent a single population age. **Dispersion (%):** Single-grain age dispersion in percent. **Central age (Ma ± 1σ):** Central fission track age in million years with one standard deviation.
**MTL (μm):** Mean track length in micrometers. **SD (μm):** Standard deviation of track length in micrometers. **N:** Number of measured track lengths. **Mean D$_{par}$ (μm):** Mean etch pit diameter (D$_{par}$) in micrometers, representing annealing characteristics of apatite.

## 4.2 Apatite fission-track dating

### 4.2.1 Methods

Apatite grains were concentrated using standard heavy-liquid and magnetic separation techniques prior to being mounted in epoxy resin on glass slides. The mounted grains were subsequently ground and polished to expose their internal surfaces. These polished mounts were etched in 5M $HNO_3$ for 20 seconds at 20 °C, following the protocols established by Gleadow et al. (2002). Apatite grains with polished surfaces parallel to prismatic crystal faces and uniform track distributions were selected for analysis. High-resolution digital images were captured in both reflected and transmitted light at a total magnification of 1000x using a Zeiss Axio Imager M1m microscope and a 3.2 MP camera. Fission-track density, confined track lengths, and etch-pit diameters ($D_{par}$) were measured (analyst: Z. Zhao) utilizing the Trackwork and Fast track systems at the Hubei Key Laboratory of Critical Zone Evolution, School of Earth Sciences, China University of Geosciences, Wuhan. The uranium content of selected grains was analysed using Laser Ablation-Inductively Coupled Plasma-Mass Spectrometry (LA-ICP-MS). Trace-element signals were normalized against $^{43}Ca$ using NIST-610 as a reference standard (Pearce et al., 1997; Vermeesch, 2017). AFT ages were calibrated using the LA-ICP-MS ζ method, with Fish Canyon Tuff apatite (28.4 ± 0.1 Ma) serving as the ζ calibration standard (Hasebe et al., 2004, 2013; Vermeesch, 2017).

### 4.2.2 Results

AFT ages for 16 samples from the Harlik Mountains span from the Early Cretaceous to the early Paleocene (111 ± 7 to 51 ± 4 Ma). Mean track lengths (MTL) were measured for 15 of these samples, ranging from 12.0 to 12.9 μm (**Table 1**). **Figure S5** presents the confined AFT length distributions for each sample. Most bedrock samples passed the chi-squared test (Galbraith and Laslett, 1993); radial plots for each sample are provided in **Figure S6**.

All but two samples (19HLK27 and 19HLK28) from the low-relief surfaces show late Early Cretaceous AFT ages (~110-100 Ma; **Table 1**; **Fig. 7**), consistent with previously published AFT and apatite (U-Th)/He dates from nearby locations (Zhu et al., 2006; Gillespie et al., 2017a; Zhang et al., 2023). The single-grain age dispersions in these samples are relatively low (see **Figure S6**), with MTLs ranging from 12.0 (n=43) to 12.8 μm (n=58), and $D_{par}$ ranging from 1.0 to 1.4 μm (**Table 1**). Two samples (19HLK27 and 19HLK28) collected from relict surface S8 yielded Late Cretaceous AFT ages of 70 ± 6 Ma and 84 ± 7 Ma, respectively. Sample 19HLK27 has a MTL of 12.6 μm (n=50), while sample 19HLK28 has a MTL of 12.4 μm (n=31). Samples from the right-lateral strike-slip fault zones dissecting the relict surfaces primarily exhibit Late Cretaceous AFT ages (~90-70 Ma; **Table 1**; **Fig. 7**). A sample collected from fault RHF 1 (23HLK02) has an AFT age of 90 ± 5 Ma, with a MTL of 12.9 μm (n=100). Samples adjacent to the RHF 2 fault zone (19HLK07/08/10-1) range in age from 73 to 84 Ma, with MTLs of 12.1 μm (n=140), 12.4 μm (n=82), and 12.6 μm (n=61), respectively. One of these samples (19HLK07) did not pass the chi-squared test (**Table 1**; **Figure S6**). Three samples (19HLK17/22/24) located within the RHF 3 fault zone record AFT ages ranging from 74 to 80 Ma, with MTLs of 12.1 μm (n=35), 12.4 μm (n=28), and 12.9 μm (n=18), respectively. A single sample (21HK04) was collected from RHF 4, yielding an AFT age of 70 ± 5 Ma.

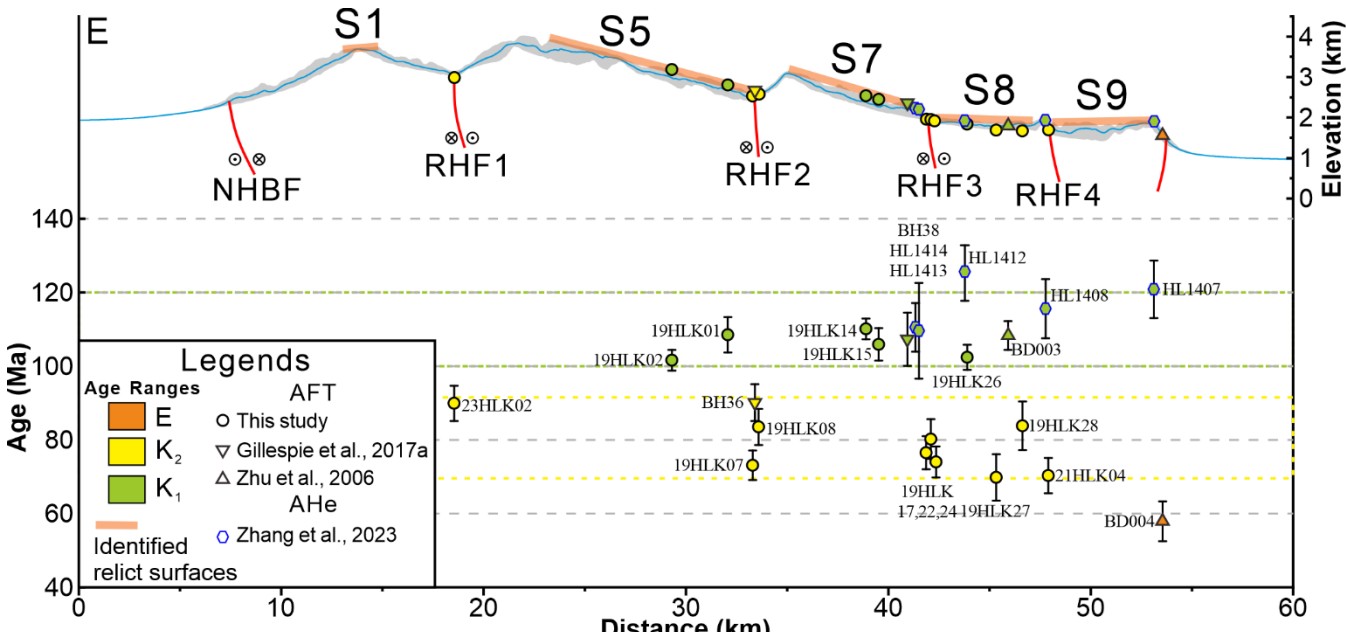

**Figure 7: AFT and (U-Th)/He data projected onto profile E-E'. AFT and (U-Th)/He data from different sources are plotted with different symbols; colours indicate ages (green: Early Cretaceous; yellow: Late Cretaceous; orange: Paleogene); see Fig. 3a for sample locations.**

## 4.3 Thermal History Modeling

### 4.3.1 Methods

To characterize the cooling history of rocks from various geomorphic units and tectonic locations, we employed inverse thermal-history modelling using QTQt software (version 5.8.0) to generate likely time-temperature (t-T) paths (Gallagher, 2012). QTQt utilizes a Bayesian Markov Chain Monte Carlo approach, using thermochronometric data and defined t-T constraints as input. The model predicts a population of acceptable thermal histories that define a posterior probability distribution. The "expected" path represents the weighted mean thermal history derived from this posterior distribution. For modelling purposes, the samples were divided into two categories: relict-surface samples (**Fig. 8**) and fault-zone samples (**Fig. 9**). Additionally, thermal histories were modelled for the two younger samples (19HLK27 and 19HLK28) collected from the low-relief surface S8, to explore potential causes for these anomalously young ages.

The AFT data were modelled using the multi-kinetic annealing model of Ketcham et al. (2007), with $D_{par}$ serving as the kinetic parameter. Inversion was constrained solely by the current surface temperature, which was set at 10 ± 10 °C. General prior ranges were established as AFT central age ± AFT central age for time and 70 ± 70℃ for temperature. The software executed 200,000 burn-in iterations followed by 200,000 post-burn-in iterations to achieve stable results. For discussion purposes, cooling rates are classified empirically into three categories: slow (< 0.5°C/Ma), moderate (0.5-2°C/Ma), and rapid (> 2°C/Ma) cooling (He et al., 2022a).

### 4.3.2 Results

The modeling result for sample 19HLK02 from surface S5 indicates a relatively steady cooling trajectory since ~130 Ma, with a phase of moderate cooling occurring between 130 and 100 Ma (~0.9 °C/Ma; **Fig. 8a**). Similarly, samples from surface S7 (19HLK14 and 19HLK15) exhibit steady cooling since ~130 Ma, with accelerated cooling rates during the 130-100 Ma interval (1.1 °C/Ma) compared to the post-100 Ma phase (0.5 °C/Ma; **Fig. 8b**).

In contrast, the thermal history of samples collected from surface S8 is more variable. Sample 19HLK26, situated away from the fault boundaries, displays a thermal history akin to that of the sample from surface S5, showing moderate cooling (0.8

°C/Ma) in the late Early Cretaceous, followed by slow cooling (< 0.5°C/Ma; **Fig. 8c**). Conversely, samples 19HLK27 and 19HLK28 record cooling starting at ~90 Ma, with a relatively rapid cooling rate of 1.9 °C/Ma until 70 Ma, which is faster than the cooling rate observed for sample 19HLK26 during the 130-100 Ma phase (**Fig. 8d**).

Overall, the QTQt modelling results for samples collected from the relict surfaces in the Harlik Mountains indicate a rapid cooling phase between 130 and 100 Ma, followed by a period of stable, slow cooling. Notably, some samples from surface S8 also experienced accelerated cooling 20-30 Ma later, around 90-70 Ma. Remote sensing reveals several minor faults intersecting these sample locations (**Fig. 6b**). All modelled thermal histories also suggest somewhat more rapid cooling from ~40 °C to surface temperatures during the last ~20 Ma.

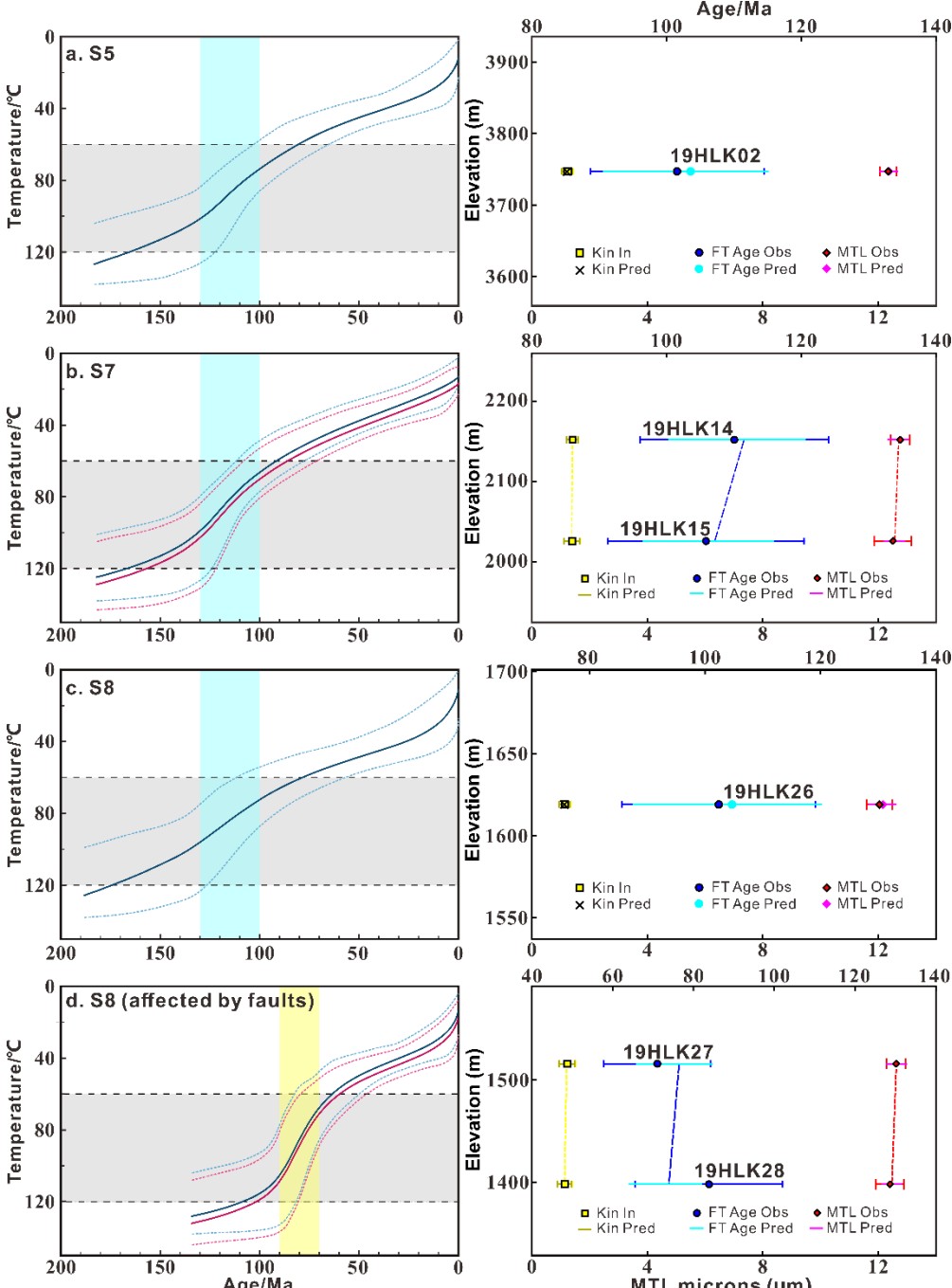

**Figure 8: Thermal history models based on AFT data for samples from relict surfaces on the southern flank of the Harlik Mountains (easternmost Tianshan) generated using QTQt software (Gallagher, 2012). Right panels show fit to the observed AFT ages and MTL.**

Samples from the fault zones RHF 2 (19HLK07) and RHF 3 (19HLK17, 19HLK22, and 19HLK24) exhibit similar t-T paths, characterized by rapid cooling during the Late Cretaceous (~90-70 Ma) at a rate of ~1.8 °C/Ma (**Fig. 9b-c**), followed by a transition to slower cooling. The thermal history of RHF 1 (23HLK02) is comparable to that of samples from RHF 2 and RHF

3, but it records a slightly earlier rapid cooling phase (2.3 °C/Ma) around 100-90 Ma (**Fig. 9a**). These thermal histories also show rapid late-stage cooling from 40-50 °C to surface temperatures during the last 10-20 Ma.

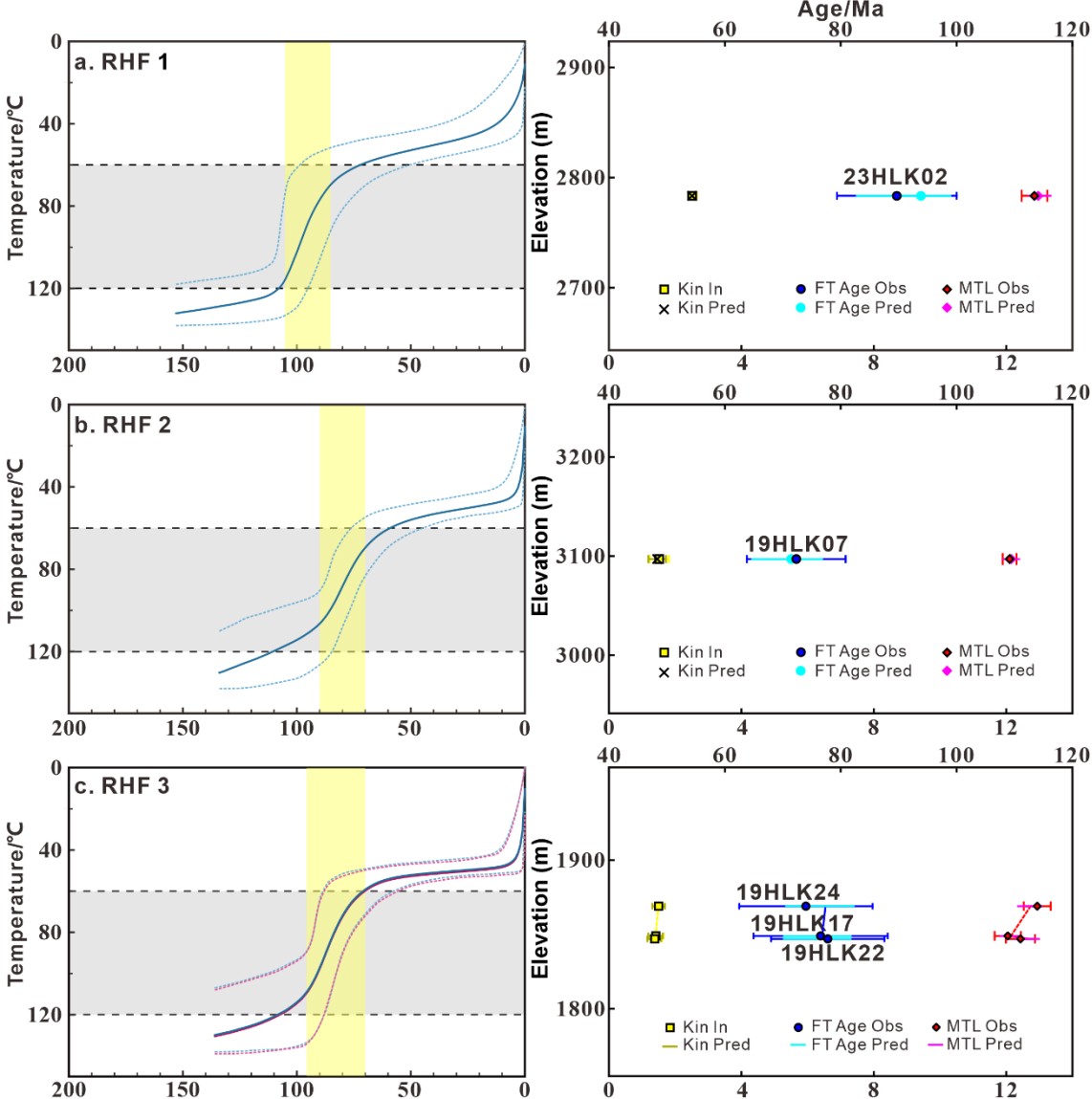

**Figure 9: Thermal history models based on AFT data for samples from fault zones on the southern flank of the Harlik Mountains (easternmost Tianshan) generated using QTQt software (Gallagher, 2012). Right panels show fit to the observed AFT ages and MTL.**

## 5. Discussion

### 5.1 Early Cretaceous exhumation of the eastern Tianshan

Thermal history models for samples from the low-relief surfaces record a relatively rapid rock-cooling and exhumation event
during the Early Cretaceous in the eastern Tianshan (**Fig. 10**). Such a thermal history contrasts to that in the Yili block (Wang et al., 2018; He et al., 2022b), Chinese Central Tianshan (Jolivet et al., 2010), and the Jueluotage Range (Sun et al., 2021), where exhumation is less pronounced during the Late Jurassic to Early Cretaceous. The low-relief surfaces in the Central Tianshan are argued to have formed during the Jurassic and to have been maintained under these relatively stable conditions (Jolivet et al., 2010; He et al., 2021b, 2022b). Cooling during the Late Jurassic to Early Cretaceous was more significant in the
Kyrgyz Tianshan compared to the Chinese western Tianshan (Sobel et al., 2006a; De Grave et al., 2012; De Pelsmaeker et al., 2015). The spatial distribution of this exhumation event in the Kyrgyz Tianshan and the Chinese western Tianshan (**Fig. 1**) suggests propagation of uplift and exhumation from the southwest towards the northeast, consistent with deformation and

exhumation driven by the collision of the Lhasa block with Eurasia during the Late Jurassic to Early Cretaceous (~160-120 Ma; Kapp et al., 2007; Ma et al., 2017; Lai et al., 2019; Hu et al., 2022).

Importantly, the available thermochronology data from the low-relief surfaces in the Harlik Mountains do not indicate any cooling events predating the Cretaceous. Evidence of rapid cooling prior to the Early Cretaceous is generally limited in the easternmost Tianshan, with a signal of Triassic cooling preserved only on the fringes of the region, outside the main fault-bound ranges (Gillespie et al., 2017a; Chen et al., 2020a). This is consistent with sedimentological evidence suggesting that the easternmost Tianshan was in an extensional setting during the Late Triassic to Middle Jurassic, characterized by slow

subsidence and relatively gentle topography, with no indication of significant uplift or exhumation during this period (Chen et al., 2019). This implies that low-relief topography likely existed in the easternmost Tianshan prior to the Early Cretaceous, but was subsequently altered by tectonic deformation and associated exhumation during this period (**Fig. 10c**).

The compressional deformation identified in the Junggar basin, easternmost Tianshan, Turpan-Hami basin, and Beishan regions during the Middle to Late Jurassic (Zhu et al., 2004; Chen et al., 2020a; Guan et al., 2021; Liu et al., 2023) was likely

influenced by the initial closure of the western Mongol-Okhotsk Ocean and Lhasa–Qiangtang collision (Zhang et al., 2019; Wang et al., 2022; Liu et al., 2023). However, low-temperature thermochronology records in regions such as the easternmost Tianshan, East Junggar, Beishan, Gobi-Altai, and Sayan-Altai predominantly record cooling during the middle Cretaceous (Jolivet et al., 2007; Glorie et al., 2012a; De Grave et al., 2014; Chen et al., 2020a, b; He et al., 2022a; Liu et al., 2023). Cooling associated with normal-fault footwall exhumation during the mid to late Early Cretaceous is recorded in the Barkol Mountains

(128-110 Ma; Chen et al., 2020b) as well as in in the Beishan area (~124-115 Ma; Liu et al., 2023). Early Cretaceous rift-related basalts (~126–99 Ma) are also common in southern Mongolia, Alxa block, and northern Qilian Shan (Graham et al., 2001; Tang et al., 2012; Hui et al., 2021). The normal faults and basalts mentioned above are considered to be associated with late-Mesozoic regional crustal extension (Chen et al., 2003; Meng, 2003; Liu et al., 2023). By this time, significant deformation associated with the Lhasa-Qiangtang collision or the closure of the Mongol-Okhotsk Ocean had ceased (**Fig. 10a**; Wang et al.,

2015; Yang et al., 2015b; Li et al., 2017; Ma et al., 2017), making it unlikely that exhumation during this period in the easternmost Tianshan can be directly attributed to far-field effects related to these orogenies. Instead, subsequent tectonic events, such as the collapse of the Mongol-Okhotsk orogen (**Fig. 10a**; ~145–100 Ma; Donskaya et al., 2008; Wang et al., 2015; Yang et al., 2015b) and the break-off of the Bangong-Nujiang Ocean slab (**Fig. 10a;** latest Early Cretaceous; Sui et al., 2013; Wu et al., 2015; Liu et al., 2018), likely played more significant roles in shaping the landscape of the study area. As the break-

off of the Bangong-Nujiang Ocean slab was later, short-lived (**Fig. 10a**), and distant from the eastern Tianshan **(Fig. 1a)**, the collapse of the Mongol-Okhotsk orogen is more likely to have provided an extensional setting for the easternmost Tianshan and adjacent areas during the mid-to-late Early Cretaceous. The relative positions of these tectonic units are illustrated in **Fig. 1a**.

On the other hand, climate change may have accelerated erosion in the eastern Tianshan and Altai regions during the mid-

Cretaceous (Hendrix et al., 1992; Pullen et al., 2020; Jepson et al., 2021a). Sedimentary records from the Junggar basin indicate a transition from arid to semi-arid conditions during the Jurassic-Early Cretaceous, evolving into seasonally wet conditions by the middle Cretaceous (Hendrix et al., 1992; Allen and Vincent, 1997; Eberth et al., 2001). Prior to ~135 Ma, the depositional environment of the Turpan-Hami Basin transitioned from an erg environment characterized by extensive eolian cross-bedded sandstones of the Liushuquan Formation to a fluvial-lacustrine setting represented by the Dahaidao Formation (Wang et al.,

2017; Zheng et al., 2023; Zhang et al., 2024). Although positive topography may have developed in the easternmost Tianshan by the Middle to Late Jurassic, the arid climate may have limited significant erosion. Conversely, the more humid climate of the middle Cretaceous likely could have intensified erosion, contributing to fluvial-lacustrine sedimentation in the surrounding basins.

**5.2 Local fault activity during the Late Cretaceous to early Paleogene**

Our geomorphological and structural analysis reveals that several right-lateral strike-slip faults segment the relict surfaces on the southern flank of the Harlik Mountains (**Fig. 3**). AFT ages from samples collected within these fault zones are mainly Late Cretaceous (~90-70 Ma), notably younger than the cooling ages of >100 Ma recorded on the surrounding low-relief surfaces. Rock cooling generally results from either erosional or tectonic (normal-faulting controlled) exhumation, or from thermal disturbance. Our observations show no significant differential erosion between these fault zones and the surrounding regions.

Given that the faults are classified as right-lateral strike-slip rather than normal faults, the most plausible explanation for the differing cooling ages is thermal disturbance within the fault zones. Microstructural analysis of the rocks within these zones provides evidence for the presence of siliceous and carbonate fluids, indicating that these rocks were influenced by hydrothermal activity associated with faulting (**Figs. 5j-l**). This hydrothermal activity may have reset the AFT system in these rocks (**Fig. 10e**). Consequently, the observed cooling ages establish a minimum timing for fault activity, suggesting that

faulting occurred during the 90-70 Ma interval. Concurrently, the cooling rate of the rocks on the relict low-relief surfaces began to slow down, indicating that regional uplift and erosion were gradually diminishing (**Fig. 10d**). During this time, deformation in the eastern Tianshan was primarily governed by these right-lateral strike-slip faults.

Throughout the Late Cretaceous, both the southern and northern tectonic belts of the Tianshan exhibited relatively low levels of tectonic activity. Consequently, many of the relict low-relief surfaces in the Tianshan region are believed to have formed

during this period (Sobel et al., 2006b; Chen et al., 2018, 2020b). While cooling events during the Late Cretaceous are less frequently recorded than those from the Early Cretaceous, they are still widely reported across Central Asia (Sobel et al., 2006a; Jolivet et al., 2010; Glorie et al., 2012a, b; De Grave et al., 2013, 2014; De Pelsmaeker et al., 2015; Chen et al., 2020a). In the Tethys tectonic domain, the collision and amalgamation of island arcs like Kohistan, Dras, and Ladakh (**Fig. 10a**; Yuan et al., 2021; Andjic et al., 2022) are thought to have generated a compressive stress field, reactivating inherited Paleozoic

structures within the western Tianshan (Schwab et al., 2004; Jolivet et al., 2010; Chen et al., 2018). Field evidence, including fault gouge zones, overprinted mylonitic fabrics, and reactivated thrust faults identified in the Narat and Bayanbulak regions, suggests multiple episodes of reactivation affecting pre-existing Paleozoic faults (Jolivet et al., 2010; Chen et al., 2018). Thermochronological constraints from AFT and apatite (U-Th)/He dating reveal significant exhumation and renewed faulting during the Late Cretaceous to early Paleocene (65–60 Ma), with deformation persisting into the Cenozoic (Jolivet et al., 2010;

Chen et al., 2018). During the Late Cretaceous and early Paleogene, with the exception of these localized fault zones, the broader western Tianshan experienced slow cooling and tectonic quiescence, facilitating the development of widespread planation surfaces (e.g., Bazhenov, 1993; Burbank et al., 1999; Glorie et al., 2010).

While the collapse of the Mongol-Okhotsk Orogen has been proposed as a potential driver of extension during the mid-to-late Early Cretaceous, some studies suggest that its influence persisted into the Late Cretaceous (van der Beek et al., 1996; Jolivet

et al., 2009; Glorie et al., 2012a; De Grave et al., 2014), possibly contributing to fault activation in the Harlik Mountains at 90–70 Ma. Evidence for Late-Cretaceous rock-cooling within a northeast-southwest extensional context has been reported in the eastern Tianshan (Song et al., 2023), Altai (Glorie et al., 2012b), Siberian Altai (Glorie et al., 2012a), Sayan (De Grave et al., 2014) and Beishan (Liu et al., 2023; Wang et al., 2024), bearing witness to widespread Late-Cretaceous fault reactivation (**Fig. 1b**). The recorded fault activity in the Harlik Mountains during 90-70 Ma likely represents similar fault reactivation. A

comparison of the tectonic setting in the northern (Glorie et al., 2012a; De Grave et al., 2014) and southern (Yuan et al., 2021; Andjic et al., 2022) tectonic belts of the Tianshan suggests that Late Cretaceous fault (re)activation in the easternmost Tianshan was likely associated with extensional tectonics resulting from the collapse of the Mongol-Okhotsk Orogen. The lithospheric adjustments following the collapse of the Mongol-Okhotsk Orogen triggered a large-scale reorganization of regional stress across NE Asia (Zhang et al., 2019), facilitating widespread crustal extension that may have extended to regions far beyond

the suture zone, including the eastern Tianshan (Song et al., 2023). Additionally, mantle upwelling related to the post-collisional extensional regime could have further contributed to the observed tectonic reactivation (Zhang, 2014).

The Tianshan-Altai-Sayan region also experienced wetter climatic conditions during the mid-Cretaceous, followed by a drier climate from the Late Cretaceous to the present (Jepson et al., 2021a). This shift to relatively arid conditions in the study area would have restricted the extent of exhumation during this period, facilitating the preservation of low-relief relict landscapes and allowing older thermochronological cooling ages to be retained.

**5.3 Cenozoic Reactivation and Fluvial Incision of the eastern Tianshan**

During the period from 90 to 70 Ma, the active right-lateral strike-slip faults in the eastern Tianshan did not induce significant differential exhumation, and rapid exhumation ceased across most areas during that time. Thermal modeling indicates a marked slowdown in cooling rate for low-relief surface samples after 100 Ma, likely coinciding with the onset of planation.

During the Cenozoic, activity in the Mongol-Okhotsk tectonic belt largely waned. To the south of the Tianshan, the collision between the southwestern margin of the Eurasian and Indian continents led to multi-stage uplift and expansion of the Qinghai-Tibetan Plateau (**Fig. 10a**; Wang et al., 2011; van Hinsbergen et al., 2012; Ding et al., 2022; Suo et al., 2022). The Cenozoic India-Eurasia collision caused significant deformation in the Tianshan and surrounding regions, reactivating pre-existing structures formed during the Paleozoic amalgamation and subsequent intra-continental deformation (Molnar and Tapponnier, 1975; Burchfiel et al., 1999). Low-temperature thermochronology analyses indicate that recent and ongoing deformation and reactivation in the Tianshan began in the Oligocene-Miocene (De Grave et al., 2007; Chen et al., 2022; Wang et al., 2023b; Jiang et al., 2024), marking the formation of the modern Tianshan topography. In the eastern Tianshan, the left-lateral strike-slip Gobi-Tianshan Fault was reported to be active during the Cenozoic (Cunningham et al., 2003), although direct age constraints for its Cenozoic activity are lacking. This fault developed into a positive flower system in the Harlik Mountains, which function as a restraining bend for the left-lateral strike-slip fault (Cunningham et al., 2003; Cunningham, 2007). This fault system facilitated exhumation of the Harlik Mountains and tilting of previously formed relict surfaces (Cunningham et al., 2003; Cunningham, 2007; Gillespie et al., 2017a). The accelerted cooling identified after ~30 Ma in our thermal models likely corresponds to this late-stage exhumation phase. Moreover, following a period of non-deposition from the Late Cretaceous to the Eocene (Shen et al., 2020), the Turpan-Hami and Barkol basins began receiving sediments again during the Oligocene to Miocene (Chen et al., 2019; Ding et al., 2024). This renewed sedimentation indicates that the Harlik Mountains have been more actively eroding since that time, marking the conclusion of relict surface development and initiating a new phase of landscape evolution in the region (**Fig. 10f**).

Recent apatite (U-Th)/He (AHe) data from the erosion surfaces of the southern Harlik Mountains (Zhang et al., 2023) show similar late Early Cretaceous ages as our AFT data, supporting rapid cooling during this time interval. Moreover, these data limit the amount of Cenozoic exhumation in the Harlik Mountains to less than 2 km, assuming a thermal gradient of 25 °C/km, a surface temperature of 20 °C, and a closure temperature of ~70 °C for AHe. Shen et al. (2024) have recently documented that the Gobi-Tianshan Fault (GTF) is active in the Quaternary, with left-lateral slip rates of 1.1–1.4 mm/yr and approximately 0.7 mm/yr of north–south crustal shortening across the Harlik Mountains. Although no direct thermochronological evidence currently confirms Cenozoic tectonic activity, knickpoint analysis identifies a series of structural knickpoints along right-lateral strike-slip faults on the southern slope of the Harlik Mountains, suggesting that these faults were active during the Cenozoic (**Fig. 4**). A limited number of Oligocene AHe ages have been reported from the Barkol Mountains, potentially linked to tectonic activity, including fault reactivation (Zhang et al., 2023). Therefore, the vertical displacement on major faults in the easternmost Tianshan may be associated with only limited erosion, similar to the western Chinese Tianshan, where recent crustal thickening and uplift has been shown to outpace erosion (Charreau et al., 2017; 2023).

The Cenozoic climate of the easternmost Tianshan, characterized by aridity as revealed by paleoclimate simulations (Jepson et al., 2021a), limited erosion and confined it primarily to river valleys and the mountain front. The $\theta_{ref}$ value for basins on the southern flank of the Harlik Mountains is ~0.22, significantly lower than the ~0.4 in the Kyrgyz Tianshan (Gong et al., 2024). This difference may result from the dominance of high-strength granite in which these valleys are incised (**Fig. 2c**) and the arid climate, both of which can reduce θ values (Kirby and Whipple, 2001; Zaprowski et al., 2005). Additionally, transient knickpoints on the southern slope of the Harlik Mountains are concentrated near relict low-relief surfaces. In longitudinal profiles, they separate an upstream unadjusted segment from a steepened downstream reach with higher incision rates. These knickpoints thus appear to record slow headward propagation and incision of the drainage into the relict surfaces, which would have started ~30 Ma. Due to limited surface erosion, crustal exhumation was insufficient to induce cooling of the basement and produce younger thermochronological ages. These conditions allowed the preservation of large-scale Mesozoic low-relief surfaces in the Harlik Mountains, even though they were deformed and tilted during late Cenozoic compression. Following uplift, there is no clear evidence of drainage reorganization that would have led to the formation of in-situ low-relief surfaces.

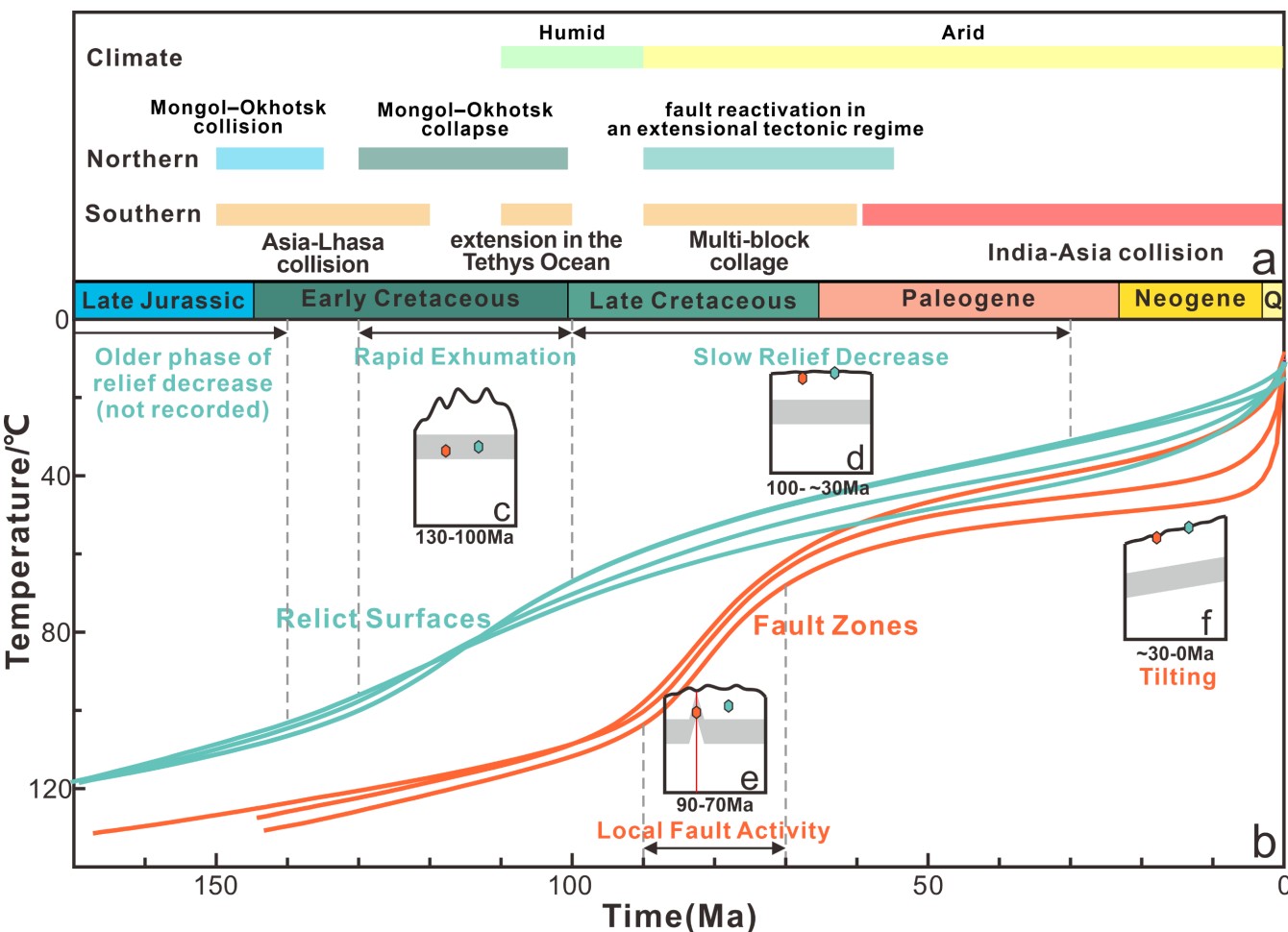

Figure 10: a. Possible driving factors for the tectonic and geomorphological evolution of the easternmost Tian Shan; b. Geomorphological and tectonic significance of the inferred exhumation history of the Harlik Mountains (time-temperature curves for the relict surfaces in green and fault zones in red); c-f. Sketches of the envisaged evolution of the relict surfaces, where the grey zone represents the partial annealing zone of apatite fission tracks, and relative location of samples located on the relict surface (green dots) and in the fault zones (red dots): c. Rapid exhumation; d. Slow relief decrease; e. Local fault activity; f. Tilting.

## 6. Conclusions

This study integrates thermochronological, structural, and geomorphological analyses to reconstruct the Mesozoic-Cenozoic landscape evolution of the Harlik Mountains in the easternmost Tianshan. Our findings reveal a complex interplay between

tectonic reactivation, faulting, and climate influences shaping the long-term geomorphic evolution of the region. The key conclusions are as follows:

1. During the **Early Cretaceous (130–100 Ma)**, rapid exhumation and rock cooling in the Harlik Mountains were primarily driven by tectonic processes associated with the collapse of the Mongol-Okhotsk orogen. Mid-Cretaceous climatic shifts, marked by increased humidity, likely enhanced erosion, contributing to basin sedimentation and modifying the regional landscape.

2. In the **Late Cretaceous (90–70 Ma)**, right-lateral strike-slip faulting, occurring under an extensional regime, segmented the relict surfaces, but did not induce significant topographic changes. Cooling patterns and fault reactivation were likely influenced by the ongoing collapse of the Mongol-Okhotsk orogen, with aridification progressively reducing erosion rates. The initiation of widespread low-relief surfaces during this period coincided with a deceleration in exhumation and cooling rates.

3. Renewed tectonic activity during the **Cenozoic (~30 Ma to present),** driven by far-field effects of the India-Eurasia collision, resulted in uplift and tilting of the Harlik Mountains, shaping the modern topography of this region. Despite active faulting and basin sedimentation, relatively low erosion rates facilitated the long-term preservation of Mesozoic low-relief surfaces. The fluvial geomorphic analysis suggests that these preserved landscapes were uplifted by late-stage faulting but not significantly altered by drainage reorganization.

While this study provides new insights, several limitations should be acknowledged:

1. **Incision Rates and Erosion Dynamics:** The lack of cosmogenic nuclide dating (e.g., $^{10}$Be exposure ages) prevents a precise assessment of incision rates and erosion magnitudes. Future studies incorporating these methods could better constrain surface processes.

2. **Climate-Tectonic Interactions:** While this study emphasizes the role of climate in erosion, a more detailed quantitative assessment, potentially through sediment provenance analysis or numerical modeling, would improve the understanding of climate-driven landscape evolution.

3. **Active Deformation Constraints:** This study primarily reconstructs Mesozoic to early Cenozoic fault reactivation based on geological and geomorphic evidence. Integrating geodetic and seismic data in future research could provide further constraints on ongoing deformation.

The Mesozoic-Cenozoic landscape evolution of the eastern Tianshan resulted from the interaction of regional tectonic (stress-field) evolution, local faulting, and climatic influences, which together shaped exhumation, deformation, and erosion patterns. The preservation of low-relief surfaces reflects the combined effects of these processes over geological timescales. This study highlights the importance of integrating geomorphic, structural, and thermochronologic data to improve understanding of landscape evolution in active orogenic systems. Future research will further refine the understanding of the complex relationships between tectonics, climate, and surface processes in the region.

**Code availability.** The QTQt software used for thermal-history modelling and inverse modelling of thermochronological data was developed by Kerry Gallagher and is publicly available at https://www.iearth.edu.au/codes/QTQt/ (Gallagher, 2012). The RadialPlotter program, used for calculating central ages of apatite fission-track data and generating radial plots, was developed by Pieter Vermeesch and is available for download at https://www.ucl.ac.uk/~ucfbpve/radialplotter/ (Vermeesch, 2009). The TopoToolbox 2 and Topographic Analysis Kit (TAK) software, used for fluvial geomorphic analysis in this study, are publicly available at https://github.com/wschwanghart/topotoolbox.git (Schwanghart and Scherler, 2014) and https://github.com/amforte/Topographic-Analysis-Kit.git (Forte and Whipple, 2019), respectively.

**Data availability.** The compilation of data is available in the Supplement.

**Supplement.** The supplement related to this article is available online.


**Author contributions.** ZZ: Conceptualization, data curation, formal analysis, investigation, methodology, visualization, fieldwork, sample collection, manuscript draft writing and revision; TS: Data curation, Investigation, Methodology, Validation, Project design, regional geology, fieldwork, sample collection, manuscript revision; GW: Project administration, funding acquisition, investigation, supervision, fieldwork, manuscript revision; PvdB: Data curation, manuscript revision; YZ: Data

curation, investigation; CM: Data curation, investigation.

**Competing interests.** The authors declare that they have no conflict of interest.

**Acknowledgements.** We sincerely thank Chengyu Zhu and Jie Wei (China University of Geosciences) for their support with

fission-track dating, as well as Wei Wang (China University of Geosciences) for assistance during fieldwork. We also appreciate Zhiyuan He (University of Potsdam) for his valuable guidance on manuscript writing, and Lingxiao Gong (University of Potsdam) for valuable suggestions on fluvial geomorphic analysis. We are grateful to one anonymous reviewer and Malte Froemchen for their insightful feedback, which greatly improved this paper. We also thank Florian Fusseis for handling this paper.


**Financial support.** This study was funded by the Key Program of the National Natural Science Foundation of China (grant No. 42430307),  the National Natural Science Foundation of China (grants No. 42172251, and No. 41972208), and Geological Survey Projects of the China Geological Survey (grants DD20179607, and DD20160060). We gratefully acknowledge the financial support from the China Scholarship Council (CSC, No. 202306410183) for funding a research stay of Z. Zhao in

Germany.

**Review statement.**

This paper was edited by Florian Fusseis and reviewed by Malte Froemchen and one anonymous referee.

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
