# Peer review of "Relict Landscape Evolution and Fault Reactivation in the eastern Tianshan: Insights from the Harlik Mountains"

_EGUsphere, 2024_

## Referee Comment (RC1)

[referee-annotated manuscript omitted]

---

## Author Response (AR1)

**Response to Reviewers**

We sincerely appreciate the thoughtful and constructive feedback on our manuscript of both reviewers. We have carefully revised the manuscript in response to these suggestions, and we summarize the key modifications below.

**Anonymous Referee:**

**Major Revisions**
**1. Incorporation of Additional Geomorphic Analyses**
To enhance our discussion on geomorphic processes and their relationship with tectonic activity, we have incorporated river knickpoint analysis, χ-maps, and longitudinal river profile analysis. These additions provide further insights into fault reactivation and landscape evolution. Additionally, we have acknowledged the limitations regarding incision rate estimation due to the lack of cosmogenic nuclide dating and have suggested it as a direction for future research.

**2. Expanded Discussion on Climate Influence**
We have strengthened our discussion on potential climate-induced erosion and sediment transport, particularly in relation to fluvial dynamics. While our primary focus remains on tectonic controls, we now provide a more detailed assessment of how climate changes may have shaped the landscape over time, along with relevant literature to support this perspective.

**3. Clarification on the Scope of Fault Data**
As our study focuses on reconstructing Mesozoic to early Cenozoic fault reactivation; we therefore primarily rely on geological and geomorphic evidence. While real-time geodetic and seismic data would provide valuable insights into contemporary fault dynamics, they fall outside the temporal scope of our analysis. We have explicitly acknowledged this limitation in the revised manuscript and highlighted the potential for future research on recent fault activity in the region.

**Minor Revisions**
In addition to addressing the major comments, we have incorporated the suggested minor revisions, including improvements to the abstract, discussion, and conclusion sections. We have also carefully reviewed the annotated PDF and made relevant modifications to enhance clarity, coherence, and readability.

**Malte Froemchen:**

**Major Revisions**

**1. Refining the Structure and Logical Flow**

To improve clarity and coherence, we have:

- Clearly separated the *Methods, Results,* and *Discussion* sections to ensure a structured and systematic presentation.
- Expanded the geomorphology and structural geology sections to provide a more comprehensive description of structural features and their significance.
- Explicitly stated the research questions in the introduction and systematically revisited them in the discussion and conclusion to enhance logical flow and reinforce key findings.

These revisions have significantly improved the manuscript's readability and organization.

**2. Incorporation of Additional Geomorphic Metrics**

To strengthen our geomorphic analysis, we have:

- Incorporated river knickpoint analysis, χ-maps, and longitudinal river profile analysis to provide a more detailed assessment of fault reactivation and landscape evolution.
- Expanded our discussion on how these selected metrics help identify tectonic uplift and active deformation zones.

While additional indices such as Surface-Roughness, Hypsometric Integral, and Elevation-Relief Ratio are valuable, we have prioritized metrics that best align with our study's objectives. We acknowledge the potential of these indices for future investigations.

**3. Climate Influence on Landscape Evolution**

We have enhanced our discussion on climate-tectonic interactions by:

- Expanding our analysis of climate-driven geomorphic processes, particularly emphasizing the role of Cretaceous aridification in preserving low-relief surfaces.
- Incorporating references to regional climatic studies to provide a broader context for our interpretations.

Although a full quantitative climate analysis remains beyond the scope of our study, we have ensured that these aspects are adequately addressed to provide a more balanced discussion.

**4. Strengthening Evidence for Fault Reactivation**

To reinforce our arguments on fault reactivation, we have:

- Provided a more detailed explanation of the paleostress analysis and kinematic indicators to improve clarity.
- Strengthened our discussion by referencing previous studies documenting similar fault reactivation processes in the Eastern Tianshan.
- Clarified how our thermochronology results align with structural observations, further supporting the interpretation of multi-phase fault activity.

These refinements ensure a more comprehensive and well-supported discussion of fault reactivation while maintaining the study's original scope.

**Minor Revisions**

In addition to addressing the major comments, we have implemented all suggested minor revisions, including:

- Restructuring the abstract for improved clarity.
- Refining figure captions for better readability.
- Enhancing specific sections to improve clarity and coherence.

**Final Statement**

Once again, we sincerely appreciate the detailed and constructive feedback of both reviewers. These comments have significantly contributed to improving our study, and we are confident that the revised manuscript now better reflects the clarity, coherence, and impact expected in this field.

**Best regards,**
**Tianyi Shen**
*On behalf of the co-authors*